# Stable yet dynamic: A cross-era comparative case study of drought impacts and social responses in Germany and Jing-Jin-Ji Region (China)

Diyang Zhang[1], Rüdiger Glaser[1], Michael Kahle[1]

[1]Physical Geography, Institute of Environmental Social Science and Geography, Faculty of Environment and Natural Resources, University of Freiburg, Stefan-Meier-Straße 76, 79104 Freiburg, Germany

*Correspondence to*: Diyang Zhang (diyang.zhang@geographie.uni-freiburg.de)

**Abstract.** Droughts, as one of the costliest weather-related disasters, have been and will continue to be part of the common human experience. However, insufficient endeavors have been made to explore drought-society
interactions in different natural and sociocultural contexts. In light of this, the present study focused on Germany and the Jing-Jin-Ji Region (China), where are dominated by marine climate and monsoon climate, respectively, and examined the similarities and differences among six extreme drought events that occurred at different episodes of the transformation from agrarian to modern societies over the last two centuries. Based on area-specific reconstructions of dry-wet indices and multilingual written documents, a comparable profile of each event was
first created under a common impact-response structural framework that encompassed five drought categories and five response attributes. Then, cross-event comparisons were conducted, highlighting the stable and dynamic elements of drought effects/impacts and social response patterns. It was found that: (1) Abnormal dry and hot conditions, vegetation damage, unsatisfactory crop performance, insufficient river flow, food insecurity, and social instability were effects and impacts independent of climate systems and were well documented by different
societies regardless of severity. (2) Despite distinct socio-environmental contexts and different disaster relief modes (e.g., top-down or bottom-up), maintaining or restoring goods supply-demand balance was an underlying logic of drought mitigation shared by different societies. Under this logic, actions often focused on the socioeconomic systems in drought-stricken areas, and the participation of governments was common due to the need for organization and coordination. (3) The diversification of documented drought impacts on socioeconomic
systems was observed in both study areas as society developed, owing to increasingly complicated economic sectors and the wider range of social concerns. (4) In recent droughts, both study areas have averted survival-threatening food insecurity through early interventions in agricultural production, multiple remedies after harvest failure, and efficient resource distribution at the national or larger scale. However, current responses have not been enough to eliminate the threat of compound drought-heatwave events to individual survival with regard to water
security (i.e., insufficient drinking water) and health (i.e., heat-related deaths). The results not only provided empirical evidence of climate-environment-society nexus that goes beyond period-specific experiences but also demonstrated the feasibility of documentary-based cross-regional comparisons in spite of linguistic differences.

## 1 Introduction

Drought, as a climate extreme that can lead to water shortages, is thought to have the largest adverse impacts of all weather-related natural hazards throughout human history, with far-reaching effects documented on each

inhabited continent (Kchouk et al., 2022; Trnka et al., 2018). As a recurring feature of all climates, a drought usually begins with a period of inadequate precipitation and/or increased atmospheric water demand and then spreads to almost all sectors of social-ecological systems as direct and indirect impacts accumulate (Cammalleri et al., 2023; UNDRR, 2021). In history, numerous cases evidenced the destructive power of drought, which resulted in water scarcity, agrarian crises, economic depression, migration, conflicts, institutional failure, regime collapse, and even the demise of civilizations (Camenisch et al., 2020; Hakenbeck and Büntgen, 2022; Hornbeck, 2023; Kennett et al., 2012; Pribyl et al., 2019; Watanabe et al., 2019; Zheng et al., 2014). In the recent past, drought-related events contributed to 34% of disaster-related deaths during 1970–2019, although they only accounted for 7% of all the disaster-related events (WMO, 2021). In the near future, anthropogenic climate change is projected to increase the likelihood and severity of droughts in many parts of the world, leading to growing population and natural resource exposures and increasing negative outcomes across all economic sectors (Elkouk et al., 2022; Runde et al., 2022).

Motivated by the growing recognition that both natural hazards and human responses can incur risks (IPCC, 2022), drought studies exhibit increasing interests in exploring human-nature interactions by analyzing meteorological processes, natural and socioeconomic consequences, and ensuing social responses based on a variety of documentary evidence (Brázdil et al., 2018; Savelli et al., 2022). Of these studies, many focused on a single event, reconstructed its course in detail, identified the cascading effects of drought across socio-ecological systems, and depicted how society addressed them at the moment (Chen et al., 2022; Gergis et al., 2010; Metzger and Jacob-Rousseau, 2020; Van Der Schrier et al., 2021; Zhai et al., 2020). Some revolved around a specific topic, such as famine (Seyf, 2010), epidemics (Burns et al., 2014), migration (McLeman et al., 2022), and conflicts (Elkouk et al., 2022; Klein et al., 2018), and discussed their nexus with drought. A few took a long-term perspective, investigated a series of drought cases in a particular place, and retraced drought impacts and social responses over time (Erfurt et al., 2019; Moerman, 2024; Noone et al., 2017). The abundant studies have shown the disruptive and complex nature of drought, enriched the understanding of drought propagation that take human actions into account, and demonstrated the contributions that historical research could make, such as providing baseline and contextualization for climate change adaptation options (Adamson et al., 2018).

However, little is known about the similarities and differences between individuals or groups in different socio-environmental contexts when facing the same climatic stimulus (i.e., precipitation deficits), as studies on past drought impacts tend to emphasize place-specific (e.g., a given region or country) experiences due to the geographical and chronological specialization of historical research. In addition, rare attention is paid to investigating the commonalities, or rather the stable elements among drought events, which have the potential to serve as a bridge between the past and present. These, to some extent, hinder the comprehensive understanding of the general mechanism through which human societies interplay with natural systems during droughts, thereby limiting the potential for identifying and sharing effective mitigation or adaptation strategies that go beyond place- and period-specific experiences.

This study aims to close the abovementioned gaps by attempting comparisons of impacts and social responses to drought in Germany and the Jing-Jin-Ji Region (China), where both have rich written documents but differ in climate systems, sociocultural backgrounds, and languages. To achieve this purpose, this study first selected three

extreme drought events in each study area, representing drought scenarios in agrarian societies, during industrialization, and in recent years. Then, a common structural framework, which comprised five categories of drought effects and impacts and five attributes of social response, was developed to help extract and integrate textual information on selected drought events from various types, forms, and languages of written documents. Next, the progression of each drought event was established under the same structural framework to make multiple drought cases comparable. Finally, cross-time and cross-regional comparisons were conducted among the six events, with a focus on the stable and dynamic elements of impacts and social responses to drought. The results are expected to provide empirical evidence of drought risks and coping strategies at different episodes of the transformation from agrarian to modern societies over the past two centuries and, meanwhile, demonstrate the feasibility of cross-regional comparisons on climate-society interactions involving various environmental conditions, sociocultural circumstances, and actors based on written documents in different languages.

## 2 Study areas

The contemporary territory of Germany covers an areas of $35.7 \times 10^4$ km$^2$, 90% of which is covered by the Elbe, Weser, Ems, Rhine, and Danube River basins (Huang et al., 2013) (Fig. 1). Forests, grassland, and arable land account for 30%, 13%, and 33% of the overall territory, respectively, and the latter two are both used for agricultural purposes. With regard to cereal farming alone, which is generally dominated by sowing in spring and autumn and harvesting since July, 57% of its yields are used as fodder. This confirms animal husbandry as a mainstay of Germany's agricultural sector (BMEL, 2020). Germany is characterized by a humid temperate climate with warm summers (i.e., Cfb class in Köppen-Geiger climate classification) (Kottek et al., 2006). Due to the maritime influence, it has a relatively mild climate compared to other areas at similar latitudes, as evidenced by a relatively narrow annual amplitude of temperature (16–25 ℃) and sufficient annual precipitation (around 800 mm) evenly distributed in four seasons. However, obvious regional differences in precipitation and groundwater recharge exist since continentality gradually increases from the northwest to the east. This puts the eastern part of Germany, such as the Elbe River Basin, under higher water stress in the summer, when evapotranspiration is at its highest and the water level of rivers is relatively low (Gebhardt et al., 2013; Glaser et al., 2007).

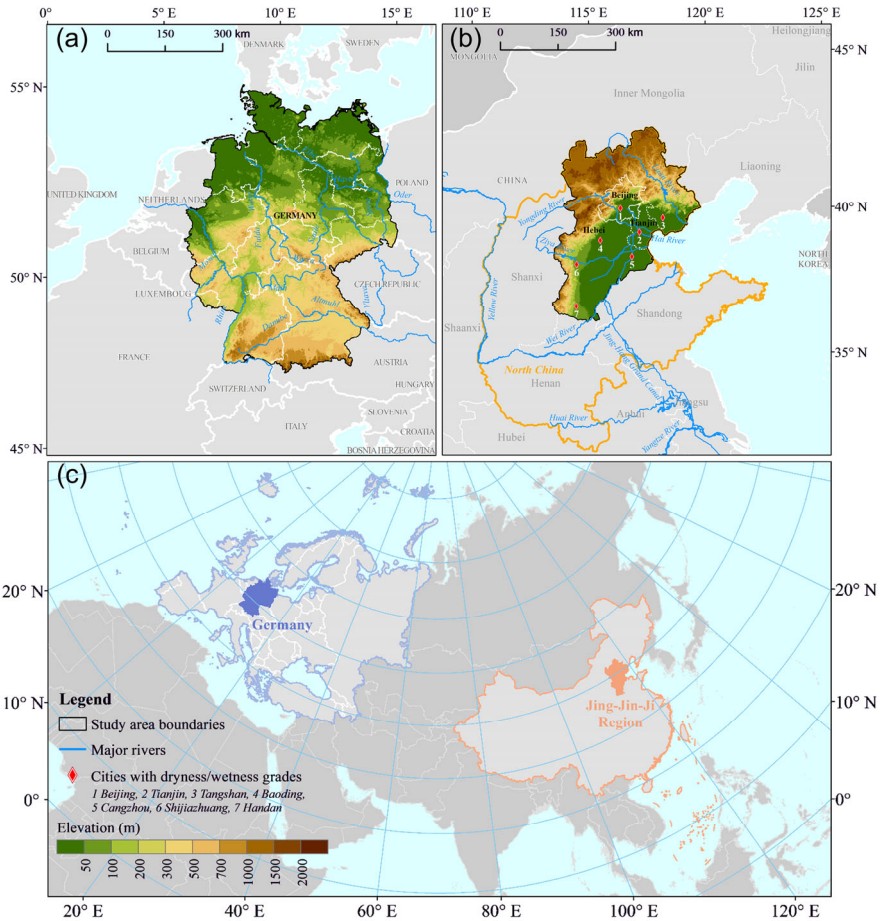

**Figure 1 Study areas: (a) Germany, (b) the Jing-Jin-Ji Region, and (c) their relative locations on the Eurasian continent**

The Jing-Jin-Ji Region includes three present province-level administrative divisions of China, namely Beijing municipality (abbr. Jing), Tianjin municipality (abbr. Jin), and Hebei province (abbr. Ji), and covers an area of $21.6 \times 10^4$ km². It is located in the northern part of North China, a traditional agricultural region and densely populated zone of China, and is mostly covered by the Hai River basin (Guo et al., 2019) (Fig. 1). Arable land accounts for 35% of the whole region and occupies most of the plains, while forests and grassland are mainly distributed in the mountainous areas and account for 35% and 9%, respectively (National Bureau of Statistics of China, 2023). Crop cultivation is the traditional livelihood of farmers in this region. Three-quarters of the annual yields come from crops sown in spring and harvested in autumn, and crops sown in autumn and harvested in the following summer contribute to the remaining quarter (Xiao, 2020). Most of this region has a continental climate with dry winters and hot summers (i.e., Dwa class in Köppen-Geiger climate classification) (Kottek et al., 2006). As deeply influenced by the East Asian Monsoon, climate here is characterized by a relatively large annual amplitude of temperature (30–32 °C), annual precipitation (400–800 mm) concentrated mostly in hot months (i.e., June to September), and four distinct seasons. During March to May, the rapid warming, months-long rainless period, and frequent windy days often lead to high evapotranspiration, insufficient surface runoff, and low groundwater levels and result in severe water stress in the spring. Additionally, the onset of the wet season varies from April to early August in different years, which depends on the strength of the summer monsoon, thus increasing the risk of consecutive seasons of drought (Wei et al., 2016; Zhao et al., 2015).

## 3 Material and methods

 ### 3.1 Data sources

In the context of climate change, the interactions among climate, ecosystems, and human society are not only producers of emerging risks but also providers of future opportunities. However, the complex nature of climate risk, the varying exposure and vulnerability of affected socio-ecological systems, and the diverse human responses make it difficult to capture those interactions by using a single metric (IPCC, 2022). Therefore, multiple sources of data are desired for gaining comprehensive and holistic insights into climate-related hazards and repercussions.

Two groups of data sources were adopted in this study (Fig. 2a). The first was (semi-) quantitative reconstructions of past dry and wet conditions, namely indices that could reflect the occurrence and extremity of precipitation deficits. Priority was given to good temporal continuity and full spatial coverage when picking the suitable index for each study area. For Germany, the annual standardized precipitation index (SPI-12 for December) from 1800 to 2022 was chosen, which was reconstructed by integrating hermeneutic information from written documents in 1500–1996 and official instrumental records from 1881 onwards (Glaser and Kahle, 2020). As for the Jing-Jin-Ji Region, annual dryness/wetness grades at seven stations (Fig. 1) were available from 1800 to 2018, which were semantic-based reconstructions and were extended with official instrumental data after 1951 (Chinese Academy of Meteorological Sciences, 1981; Weather China and National Climate Centre, 2019; Zhang et al., 2003; Zhang and Liu, 1992), and were converted into to a regional dry-wet index (Fig. 2b) by Eq. (1).

$$DW(t) = \left(\frac{m+n}{N}\right) \times \left[\sum_{i=1}^{m} \beta_5 \times A(i,t) + \sum_{i=1}^{n} \beta_4 \times A(i,t)\right] + \left(\frac{j+k}{N}\right) \times \left[\sum_{i=1}^{j} \beta_2 \times A(i,t) + \sum_{i=1}^{k} \beta_1 \times A(i,t)\right],(1)$$

Where $DW(t)$ is the dry-wet index for the whole Jing-Jin-Ji Region in year $t$. $N$ is the number of all stations in the Jing-Jin-Ji Region. $m, n, j$, and $k$ denote the number of stations reporting very dry (grade 5, frequency: about 10%), dry (grade 4, frequency: 20% to 30%), wet (grade 2, frequency: 20% to 30%), and very wet (grade 1, frequency: about 10%). $A(i,t)$ is the grade anomaly at station $i$ in year $t$. $\beta$ refers to the coefficient for each grade (i.e., $\beta_5 = -2, \beta_4 = -1, \beta_2 = 1, \beta_1 = 2$). Stations with grade 3 (normal years, frequency: 30% to 40%) and no data are excluded in the calculation, because they reflect a situation that is neither dry nor wet. The frequency of each dryness/wetness grade is from Chinese Academy of Meteorological Sciences (1981).

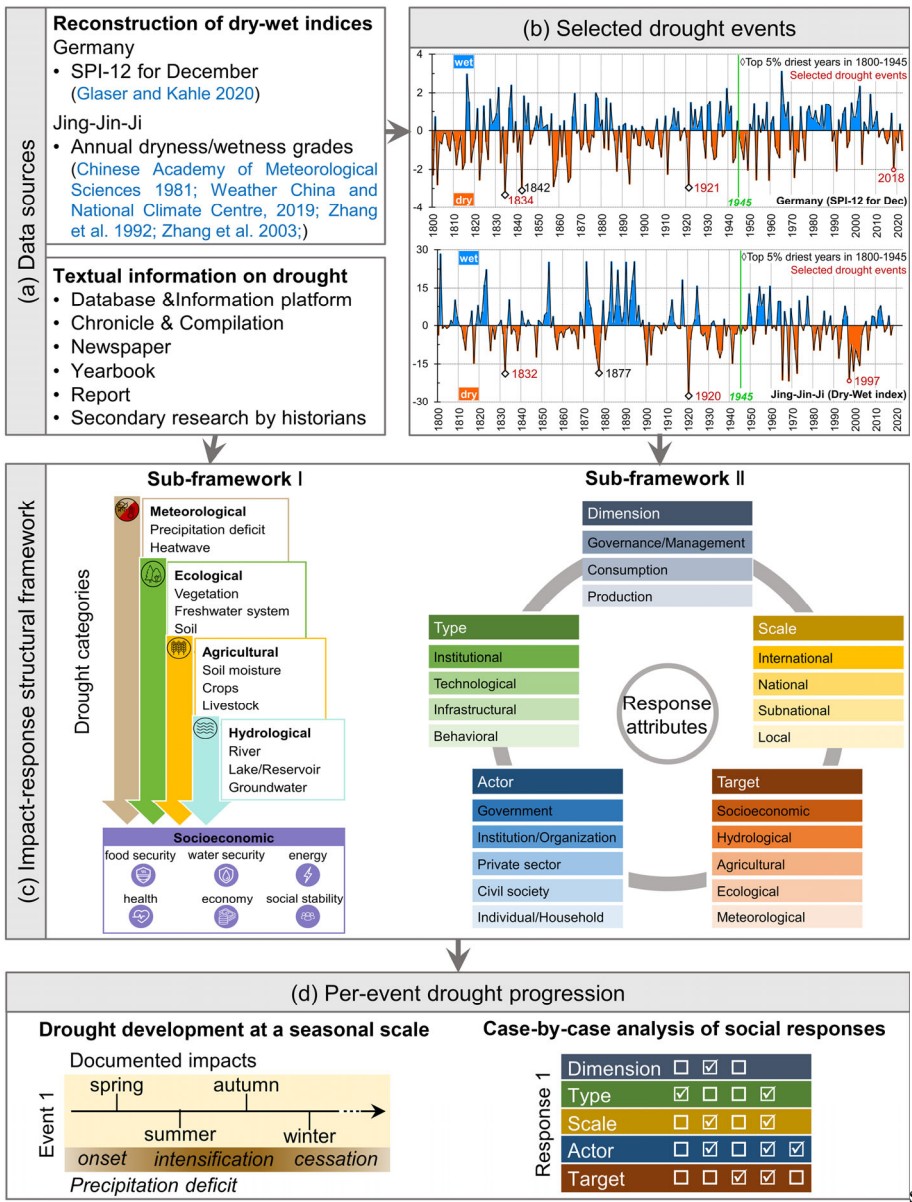

**Figure 2 Conceptual framework**

The second group of data sources were multiple types of written documents listed in Supplement A, which could provide details on the development of precipitation deficits, subsequent impacts, and actions taken by humans. The original content of drought-related descriptions was either directly obtained from subject-specific databases and information platforms or newly extracted from chronicles and compilations, local or national newspapers, official yearbooks, and formal or informal reports. For instance, specific records could be searched in the Collaborative Research Environment tambora.org (Riemann et al., 2015) and the Integrated Natural Disaster Information System of Qing Dynasty (Fang et al., 2020; Xia, 2015) by disaster type, time of occurrence, and position, while textual information on the social focus at a given moment needed to be captured manually from newspapers such as the *Freiburger Zeitung* and *People's Daily*. Meanwhile, secondary research (e.g., peer-reviewed papers, monographic studies) by historians was adopted as a supplement when the original materials were unavailable.

**3.2 Selection of drought events**

The selection of drought events was based on the extremity of dry conditions and the representativeness of different social development stages. In other words, for a selected event, its precipitation deficits should be severe enough to be considered as one of the greatest exogenous pressures on a specific region with respect to climatic conditions; meanwhile, its occurrence should be at a certain episode in the profound transformation from agrarian to modern societies that human society experienced over the last 200 years (IPCC, 2022). Following this principle, it is expected to capture the performance of socio-ecological systems under the pressure form extremely dry conditions in representative socio-environmental contexts, although the drought events ultimately selected are not necessarily the most socially disruptive ones.

Before setting the criteria for selection, a couple of differences across regions or over time should be noted, which include:

(1) There are fundamental differences between Germany and the Jing-Jin-Ji Region in terms of climate systems and usable proxies for long-term climate reconstructions. This makes it nearly impossible to simply compare the dry-wet indices available for the two study areas and select events with close meteorological features.

(2) As dry-wet indices for both study areas absorbed official instrumental data at certain points, uncertainties introduced by non-climatic influences on weather stations are also noteworthy, such as possible changes in standard procedures, observation techniques, network densities, and station locations over time (Brugnara et al., 2020, 2022). This challenges the identification of events with the same intensity of precipitation deficits at different periods within one study area.

(3) The pace of social transformations and consequent environmental changes differed from time to time, which have accelerated considerably after the Second World War, i.e., in the post-1945 period (McNeill and Engelke, 2014). This hints at the need for developing selection criteria segmentally.

In other words, based on currently available data, it is impracticable to ensure that the two study areas were always under the same strong exogenous pressures from precipitation deficits in selected drought events. Thus, this study turned its attention to the most significant deviations (i.e., the driest years) reflected in the dry-wet indices applied to each study area and then set criteria independent of the location and data. Additionally, distinctive selection criteria were applied for periods before and after 1945:

(1) For the period from 1800 to 1945, in each study area, all years with a SPI-12 or dry-wet index < 0 were first identified as dry years. Then, those dry years were ranked according to the specific value of the index. Next, the top 5% driest of all dry years were picked out, and events covering those extremely dry years were considered as potential options. After that, two events were selected among them to represent drought in agrarian society and during industrialization, respectively.

(2) For the period after 1945, the most recent drought events with prolonged and severe precipitation deficits were selected for each study area, aiming to best reflect the context of modern society.

Ultimately, three pairs of drought events were selected (Fig. 2b), namely the Germany 1834 (SPI-12 = −3.33) and Jing-Jin-Ji 1832 (dry-wet index = −18.81) events in agrarian societies, the Germany 1921 (SPI-12 = −2.95) and Jing-Jin-Ji 1920 (dry-wet index = −27.58) events during industrialization, and the Germany 2018 (SPI-12 = −2.02) and Jing-Jin-Ji 1997 (dry-wet index = −21.68) events in modern societies. These events were named after the driest

years they witnessed, and their durations were determined as described in Sect. 3.4. Admittedly, the temporal distance of the two modern drought events was greater than that of the selected paired drought events during the period of 1800–1945, as the dry and wet conditions in the two study areas showed different trends, especially in the past three decades (Fig. 2b). In this situation, priority was given to the severity and recentness of dry conditions when extremity, modernity, and simultaneity were too difficult to achieve at the same time. It should also be noted that the relatively well-developed railway system was an important consideration for dividing the Jing-Jin-Ji 1920 event into the era of industrialization, although the broad population shift from agriculture to manufacturing had not yet been observed in this region at the time (Li, 2007; Wu, 1985).

**3.3 Development of impact-response structural framework**

Comparability is an important issue in documentary-based comparative studies involving different areas at different periods, as the type, character, complexity, and information density of data sources are always varied (Erfurt et al., 2019). Conducting comparative analyses under a common structure reflected in all cases can be a feasible strategy for ensuring comparability, which has been applied in domains such as human response systems in the face of natural hazards (Knight, 2001). Following this strategy, this study formulated a common structural framework to guide the characterization of drought progression in multiple cases, which consists of two sub-frameworks as below.

In the sub-framework I (Fig. 2c), five drought categories were abstracted from various textual information on the negative outcomes of drought, with reference to the meteorological drought, agricultural drought, hydrological drought, groundwater drought, and socioeconomic drought defined by Bernhofer et al. (2015). To better serve documentary-based research from a historical perspective, this study created a separate category for ecological drought, merged groundwater drought into the hydrological category, and developed comprehensive sub-categories with typical effects and impacts of drought (Table 1). In addition, compound heatwaves were incorporated into the meteorological category, considering not only the contribution of abnormal high temperature to the likelihood and destructiveness of drought (Zscheischler et al., 2018), but also the not rare descriptions in written documents of the combined effects of heat and drought. Supplement B provided a list of keywords for the 56 specific effects and impacts in each drought category, along with examples of how they were documented in historical, early modern, and contemporary source materials.

**Table 1 Drought categories, sub-categories, and specific negative effects and impacts**

| Categories | Sub-categories | Effects and impacts |
| --- | --- | --- |
| Meteorological | Precipitation | Deficit, severe deficit (described by degree adverbs such as unprecedented, rare, extreme, severe, great, very, etc.) |
| | Temperature | Heatwave (abnormal hot weather lasting from several days to months or extreme maximum temperatures recurring in a short period) |
| Ecological | Vegetation [a] | Abnormal phenological period, poor plant growth, plant death, decreased vegetation cover, insect plague, wildfire |
| | Freshwater system | Increased water temperature, deteriorated water quality, fish death |
| | Soil | Dust storm |
| Agricultural | Soil moisture | Insufficient moisture, cracks in land |
| | Crop | Difficulty in sowing, growth affected, poor quality, harvest failure |

| | | |
|---|---|---|
| | Livestock | Lack of fodder, lack of bedding material, decrease in production, death |
| Hydrological | River | Low water level, reduced flow, cutoff |
| | Lake & Reservoir | Shrinking water body, decreased water stock, dry up |
| | Groundwater | Dropped water level, insufficiently derfilled aquifer, wells dry up |
| Socioeconomic | Food security | Affordable (price), available (supply), famine |
| | Water security | Drinking water (supply, quality, price), hygiene and sanitation, other usage (daily life, agriculture, industry, etc.) |
| | Energy | Fuel (cooking, heating, other usage), electricity (generation, supply, price) |
| | Health | Malnutrition and decreased labour capacity, injury (fire disaster), heat-related illness, epidemic, mental health, death |
| | Economy [a] | Private property losses (fire disaster), infrastructure damage, decrease in income, increase in expenditure, obstruction of transport, limitations on other economic activities |
| | Social stability | Conflict, displacement and migration, crime, moral and ethical collapse |

a. The forestry industry is an important economic sector in Germany but not a major livelihood in the Jing-Jin-Ji Region. Thus, instead of setting a separate (sub-) category, this study classified drought impacts on forestry into either ecological or socioeconomic categories based on specific descriptions. For example, impacts described as trees in poor condition belong to the vegetation subcategory in the ecological category; while, impacts described as decreases in income due to damaged trees belong to the economy subcategory in the socioeconomic category.

240

The sub-framework II (Fig. 2c) depicted five general attributes of social responses to drought, based on which diverse responses could be comparable to each other after appropriate abstraction:

(1) *Dimension* demonstrates the levels of social responses. Production-level responses intend to minimize drought impacts on producing primary products by taking remedial actions based on current resources (e.g., irrigation, replanting short-season crops to compensate for harvest failures) or exploiting environmental potentials (e.g., digging wells deeper). Consumption-level responses aim to balance the supply and demand of a particular good or service by expanding supply (e.g., transporting food or water to drought-stricken areas, releasing grain reserves to the market), reducing waste and/or outflow (e.g., promoting water conservation), increasing purchasing power (e.g., increasing remuneration), restraining demands (e.g., water restrictions and rationing), adjusting allocation strategy (e.g., prioritizing domestic consumption in case of electricity shortages), and finding substitutes (e.g., eating famine food such as ground corn cobs and tree leaves). Governance/management-level responses attempt to maintain or recover the function of social-ecological systems through assessment, intervention, coordination, and social innovation, which, in many cases, are designed from a top-down and comprehensive perspective (adapted from Fang et al. (2019)).

(2) *Type* specifies the manner in which a social response is conducted. Behavioral responses involve attempts to mitigate risks or repercussions directly, such as praying for rain, irrigation, emergence slaughter, transporting life necessities to and relocating from drought-stricken areas, and temporarily relaxing or restricting regulations. Infrastructural responses refer to the investment, construction, and maintenance of equipment and infrastructure, such as firefighting equipment, water facilities, transport infrastructure, and power supply systems. Technological responses comprise implementing monitoring and early warning systems, improving farming practices and water-saving technologies, breeding and planting climate-resilient plants, providing technical assistance and training, and updating coping concepts. Institutional responses include creating long-term policies and programmes, establishing laws and regulations, and setting up permanent institutions or organizations (adapted from Berrang-Ford et al. (2021)).

(3) *Scale* characterizes the spatial extents of social responses. Actions can be taken within the affected place (i.e., local scale); engage less-affected areas, such as neighboring provinces or states (i.e., subnational scale); be supported by central authorities or have nationwide influence in present-day territories (i.e., national scale); and involve other countries and/or international organizations (i.e., international scale).

(4) *Actor* indicates who undertakes social responses. It ranges from individuals or households to civil society, the private sector, institutions or organizations, and multiple levels of government, indicating the extent of social involvement in coping with droughts.

(5) *Target* identifies the categories of drought effects and impacts (i.e., from meteorological phenomena to socioeconomic impacts) that a given social response aims to address. This attribute was designed to show the

key concerns of each social response reflected in their original intentions, whether or not those intentions were achieved in the end.

**3.4 Establishment of the drought progression for each selected event**

Providing comprehensive and comparative profiles of selected drought events (i.e., drought progression) was the first step in comparing the drought-society interactions they implied. This study took the onset and cessation of

precipitation deficits as the beginning and end of an event, which was a trade-off between the limited availability and relatively narrow scope of drought records in history and the abundant and comprehensive drought reports at present. As sequels to the lack of precipitation, other effects and impacts of drought may not take place (e.g., electricity shortage) or be well documented (e.g., degree of soil moisture at different soil depths) in every region and at every episode of social development. In addition, some impacts can continue in an inconspicuous manner

over a non-measurable span after precipitation returned to normal (e.g., reduced coping capacity for future hazards), which goes more or less beyond the scope of the current case study. Nevertheless, it should be noticed that this definition of event duration is not ideal for exploring drought effects in the long run (e.g., post-drought labor shortages due to climate exodus) or drawing a complete picture of drought resilience over a period.

Subsequently, drought progression of the six selected events was established, respectively, under the guidance of the abovementioned structural framework (Fig. 2c, 2d). In brief, when dealing with historical evidence about a particular event derived from multiple types of written documents, categorization was employed to identify specific effects and impacts of drought, while different records of the same response action were integrated into one way of responding. Compared with counting the number of mentions in written documents, these approaches

could reduce the influence of changes in the availability and nature of historical evidence over time on per-event analysis and cross-event comparison. For example, water insecurity in the summer was considered to have occurred only once, whether there were two records of lacking drinking water in a summer in history or ten reports on difficult drinking water supply in a summer in recent years. Similarly, irrigation was counted only as one way of responding, no matter how many mentions of farmers irrigating their fields in a single event. Thus, differences

in the quantity of historical evidence between the past and present would not be captured as differences in drought impacts and/or social responses when comparing historical and recent events.

To piece together a clear timeline of each selected event, different categories of drought effects and impacts were identified at a seasonal scale according to the following principles:

(1) For an impact with records detailed to the date or month, the season to which the date or month belonged was adopted.

(2) For an impact that had records with clear seasonal information, the documented season was adopted directly.

(3) For an impact described as persisting across several seasons, all seasons mentioned were taken into account.

(4) For an impact without clear seasonal information but derived from time-sensitive sources (e.g., newspaper),
the season in which those sources were published was chosen.

(5) For an impact that was only documented as existing in the event, it was omitted here.

Regarding social responses made in a particular event, they were not analyzed at a seasonal scale but on a case-by-case basis, due to their relatively vague temporal information (especially when these actions ended). In other
word, each way of responding should be marked by one or more classes from each of the five response attributes (Fig. 2d). Additionally, to better extract and visualize the patterns of social response, this study first grouped all ways of responding in a particular event according to their *Dimension* (i.e., production-level, consumption-level, or governance/management-level responses). Then, the proportion of social responses belonging to a given group was calculated by Eq. (2), illustrating the general distribution of responses at different levels during this event.
Next, within each group, the proportion of social responses marked by each class in the four other was calculated by Eq. (3), depicting the manners (*Type*), spatial extents (*Scale*), social involvement (*Actor*), and key concerns (*Target*) of responses at different levels. Finally, cross-event comparisons were conducted between the response patterns in different regions and/or at different social development stages.

$$P(k) = [N(k)/N] \times 100\%, (2)$$

$$P(m_i, k) = [N(m_i, k)/N(k)] \times 100\%, (3)$$

Where $P(k)$ represents the proportion of social responses belonging to group $k$ ($k$=production, consumption, or governance/management) during a particular event. $N(k)$ denotes to the number of social responses falling into group $k$. $N$ refers to the total amount of social responses that were mentioned in the entire event. $P(m_i, k)$ represents the proportion of social responses in group $k$ that were marked by class $i$ of attribute $m$ ($m$= type,
scale, actor, or target). $N(m_i, k)$ denotes the number of social responses in group $k$ that were marked by class $i$ of attribute $m$. The proportions of social responses marked by different classes of a given attribute could sum to more than 100%, since classes of attributes except for *Dimension* were not necessarily mutually exclusive.

Due to space limitation, the progression of selected drought events was described in relatively general terms in
Section 4, rather than itemizing historical evidence. Details on specific effects and impacts of drought and ways of responding were elaborated on event by event in Supplement C and Supplement D, separately.

## 4 Results

### 4.1 Recurring summer droughts in agrarian societies

The Germany 1834 drought event occurred two decades after the Napoleonic Wars. In this context, Germany had
undergone a series of territorial reshufflings and social reforms, which are believed to lay the foundations of the modern state and society, but had not been unified into a nation-state. Moreover, some important parts of its

traditional structure remained in place during the post-Napoleonic era, such as the nobility's local power and the irreplaceable role of agriculture among all livelihoods (Grab, 2003; Slicher van Bath, 1963). This event was marked by recurring hot summer droughts. It spanned from autumn 1833 to summer 1836 and culminated in summer 1834 with 16 specific effects and impacts of drought in 11 sub-categories (Fig. 3a, Supplement C). Of those 12 seasons, nearly three fifths were considered too dry due to insufficient precipitation, with heatwaves occurring in more than two-fifths abnormal dry seasons; a quarter observed ecological impacts in the form of damage to vegetation; close to one third saw drought affecting agriculture, manifested as difficulties in crop production and livestock feeding; and almost three fifths reported low surface water and groundwater levels in the hydrological category. Meanwhile, drought spilled over into socioeconomic systems in slightly over two fifths of the 12 seasons, and food supply risk, water insecurity, economic losses, and water conflicts depicted the challenges that society faced at the time.

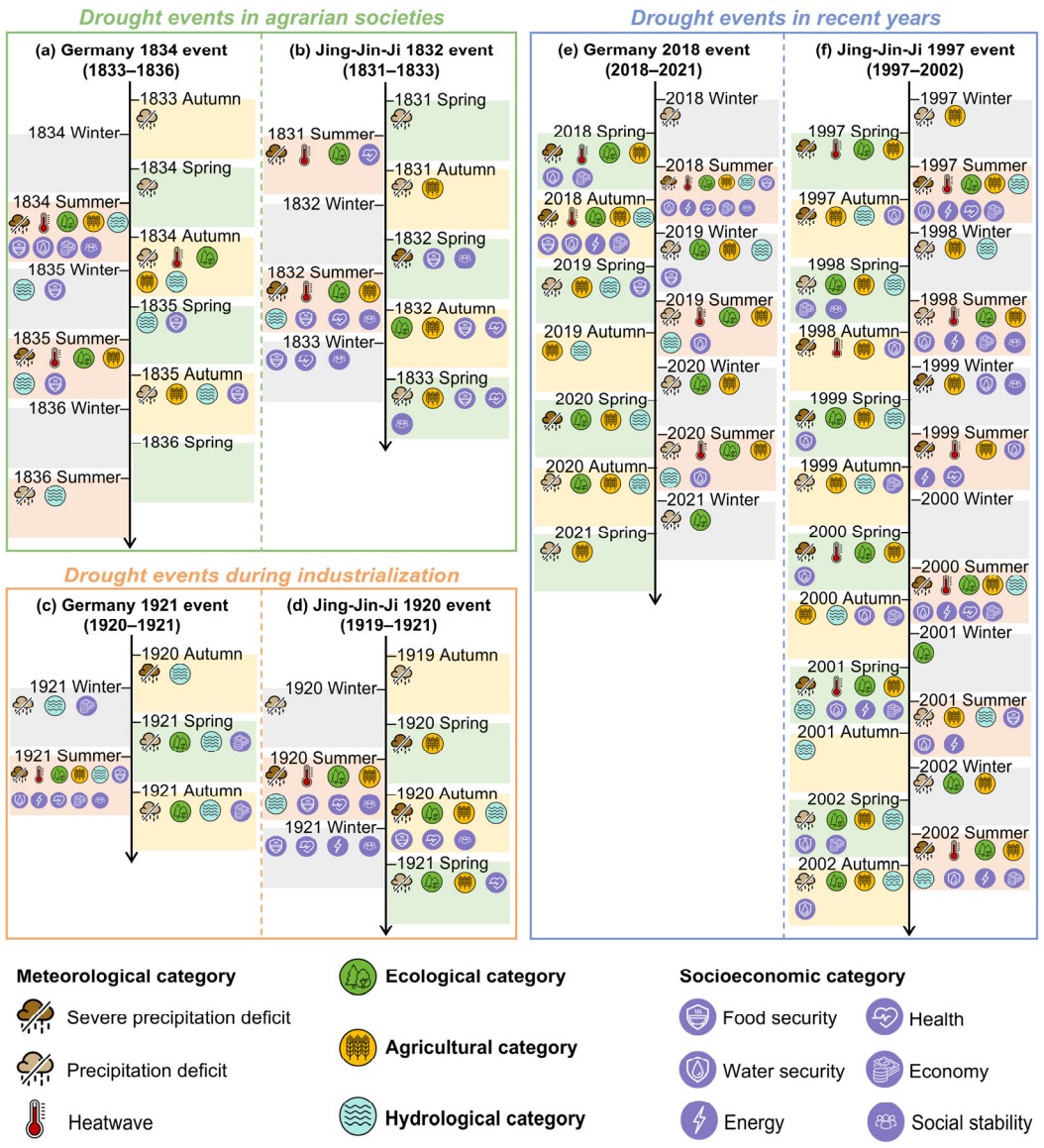

**Figure 3 Progression of extreme drought events in Germany (abbr. DE) and the Jing-Jin-Ji Region (abbr. JJJ), depicted by different (sub-) categories of drought effects and impacts at a seasonal scale. For each event, the beginning and ending seasons correspond to the first and last mentions of precipitation deficits, respectively.**

Ten ways of responding were documented in this event (Fig. 4a, Supplement D), six of which were taken at the consumption level to deal with inadequate supplies of fodder and water by expanding supply, reducing demand, or adjusting allocation strategy. At this level, government was the most engaged actor that was visible in 50% of the responses, implying the active participation of local authorities in addressing supply-demand imbalances in drought-stricken areas despite the lack of central authority. Meantime, individuals/households also took a few actions to prevent livestock from starvation, while the civil society organized water transport to alleviate water scarcity in severely affected communities. For the two responses at the production level, farmers attempted to compensate for summer cereal failures by replanting, while community wells were carefully protected to ensure water accessibility for the whole community. As for the two responses at the governance/management level, precautionary water storage in community cisterns and the activities of fire brigades were endeavors to protect local socio-ecological systems from fire threats. In short, the dashboard of this event (Fig. 4a) illustrated a major response pattern in which actions were often taken locally to mitigate drought impacts on agricultural and socioeconomic systems through maintaining or restoring the supply-demand balance of scarce goods. Private sector was the only actor not acting in this event.

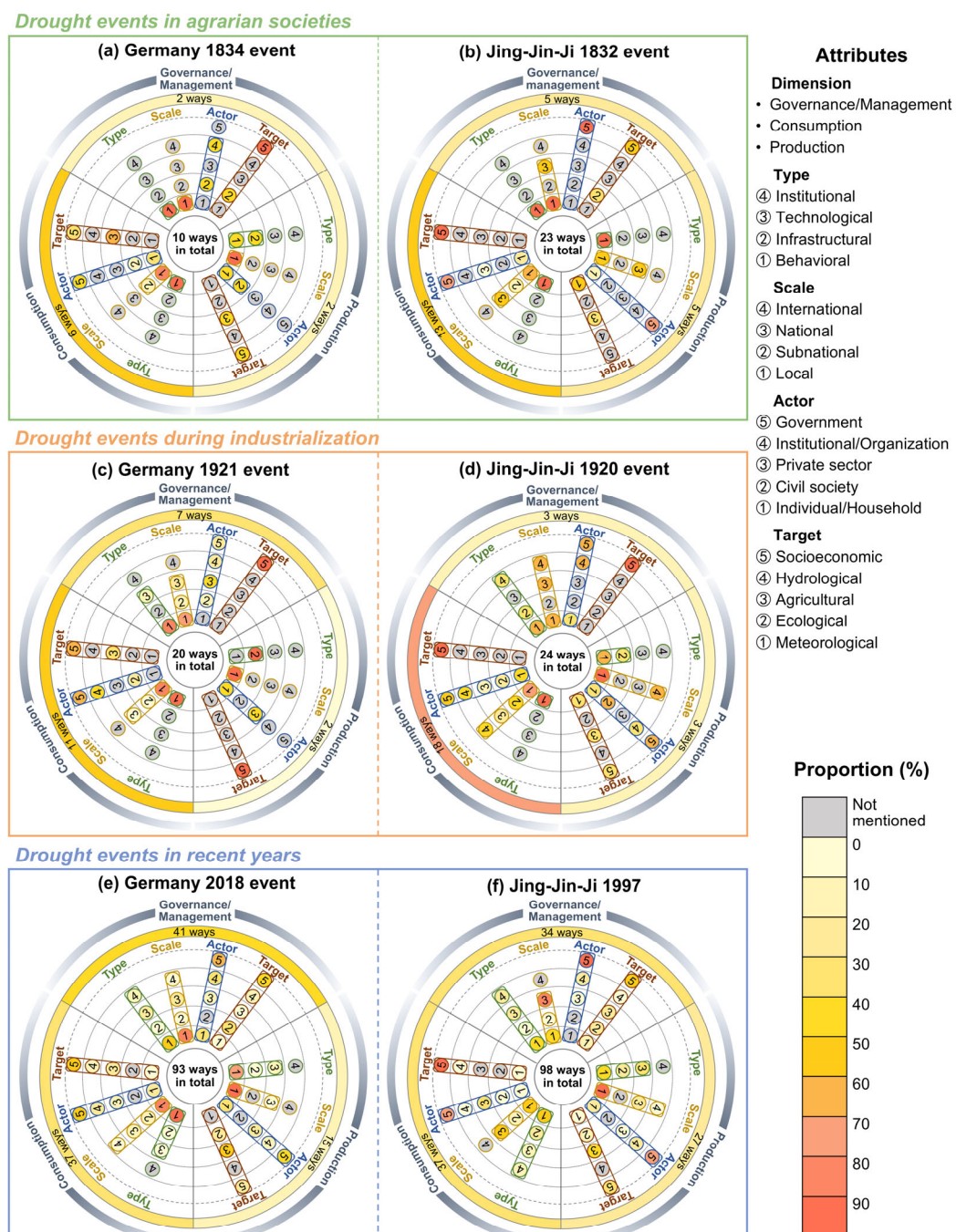

**Figure 4 Dashboards of social response patterns in selected drought events, outlining levels (*Dimension*), manners (*Type*), spatial extents (*Scale*), social involvement (*Actor*), and key concerns (*Target*) of drought coping strategies in different regions and at different social development stages. For a particular event, the proportions of responses belonging to different *Dimension* were calculated by Eq. (2) in the section 3.4. Meanwhile, the proportions of responses in a given dimension that were marked by each class of the four other attributes were calculated by and Eq. (3) in the section 3.4.**

Roughly two years earlier, recurring summer droughts compounded with heatwaves were also documented in the Jing-Jin-Ji Region, where drought-affected groups lived very close to the center of power of the Qing Dynasty and their livelihoods largely relied on crop farming. The Jing-Jin-Ji 1832 drought event began with precipitation deficits in spring 1831, peaked in summer 1832 with 13 specific effects and impacts of drought in eight sub-categories, and ended with sufficient rainfall in the early summer of 1833 (Fig. 3b, Supplement C). From spring 1831 to spring 1833, rather dry conditions prevailed two thirds of the time, with one third of all abnormal dry

seasons accompanied by heatwaves. Among the nine seasons, three mentioned ecological impacts described as vegetation damage; over four witnessed impacts on agriculture highlighted by harvest failures of crops; and one reported river cutoff, the only impact belonging to the hydrological category. Additionally, drought impacts on socioeconomic systems were recorded in two-thirds of the nine seasons, with particular concerns about food security, health, and social stability. The displacement and death of individuals due to drought-induced famine and the consequent rise in crime put considerable pressures on the society.

During this event, 23 ways of responding were observed (Fig. 4b, Supplement D). Here, responses belonging at the consumption level were the most popular (56.52%), mainly designed to mitigate drought impacts on food security through actions at local, subnational, and/or national scales. Similar to Germany, expanding supply and adjusting allocation strategy were also the main approaches adopted by both governments and individuals/households. Meantime, both production-level responses and governance/management-level responses accounted for slightly over 20%. The former were reactions of governments and individuals/households to insufficient rainfall and subsequent difficulties in local agricultural production, while the latter focused most on the assessment and control of locust plague, food insecurity, and social instability—led by the central authority and carried out by local officials. Overall, the main response pattern in this event showed similarities to that in the Germany 1843 event. In both cases, it was common practice that major efforts were put into maintaining or restoring the supply-demand balance of scarce goods, with the intention of tackling drought-induced socioeconomic issues, and governments normally took an active part in these actions. Nonetheless, different from the Germany 1834 event, this event showed a clear top-down mode in which disaster relief was directed and coordinated by the central authority and implemented in drought-stricken areas with support from less-affected neighboring regions, displayed on its dashboard (Fig. 4b) as high degree of government involvement (82.61% of all responses) and large spatial extents of social responses (about 60% at the national scale).

**4.2 Yearlong drought during industrialization**

The Germany 1921 drought event occurred in the initial years of the Weimar Republic, when the country had benefited from the technological progress of the Industrial Revolution but was also limited by the burdens of the First World War (Erfurt et al., 2019; Streb et al., 2006). Overall, below-average precipitation persisted in five consecutive seasons from autumn 1920 onwards, two fifths of which coincided with heatwaves, and summer 1921, when this event came to a peak, witnessed up to 28 specific effects and impacts of drought in 16 sub-categories (Fig. 3c, Supplement C). By the autumn of 1921, three of the five seasons had observed ecological impacts, involving threats to both vegetation and freshwater systems; one season had seen drought affecting agriculture in terms of soil moisture, crop, and livestock; all seasons had experienced at least one of the impacts in the hydrological category, i.e., low water levels in major rivers; and four of the five seasons had reported impacts on socioeconomic systems, such as economic losses. In addition, summer 1921 was described as a serious challenge for society. At that moment, troubles arose in all six sub-categories of the socioeconomic category, ranging from unaffordable food prices to widespread profiteering, some of which worsened the living conditions of vulnerable groups and led to remarkable increases in infant mortality.

Twenty ways of responding were mentioned in this event, many of which belonged to the dimensions of consumption and governance/management (Fig. 4c, Supplement D). Specifically, consumption-level responses,

comprising 55% of the total responses, sought to address problems with livestock feeding, food price, drinking water, and electricity. Over 80% of them were behavioral responses attempting to mitigate the negative repercussions at a local scale. Although the proportion of governance/management-level responses was 20 percentage points lower than that of consumption-level responses, more than 70% of them were also made in the behavioral manner and at a local scale. Socioeconomic issues were the only concerns of responses at this level, including food and water insecurity, fire disaster, rising morbidity and mortality in vulnerable groups (e.g., infants), and potato profiteering. There were only two responses at the production level, but both were made in the manner of infrastructure construction focusing on the improvement of local water availability. Similar to the previous cases in agrarian societies, balancing goods supply and demand remained a preferred strategy for German society at the time to mitigate drought impacts on local socioeconomic systems. Nevertheless, a broader social involvement was observed in this case. In brief, government (45%) and institution/organization (35%) were the most engaged actors, while private sector (20%), civil society (20%), and individual/household (5%) were also involved to some extent. In addition, national-scale responses were also mentioned in this event, but unlike in the 1832 Jing-Jin-Ji event, most of these actions were not strong governmental instructions for disaster relief.

A year earlier, the Jing-Jin-Ji Region also suffered from a long absence of rainfall. At that time, this region was witnessing the early stage of industrialization, which brought about the development of railways, but much of it was still under a self-contained natural economy where grains were produced mainly for producers' direct consumption. Additionally, the region was undergoing fractured governance by warlords after the recent fall of the Qing Dynasty (Fuller, 2013; Wu, 1985). The Jing-Jin-Ji 1920 drought event started with insufficient rainfall in autumn 1919, peaked in summer 1920 with 13 specific effects and impacts of drought in nine sub-categories, and ended after a brief return of precipitation deficits in spring 1921 (Fig. 3d, Supplement C). Of the seven seasons, six were identified as abnormal dry seasons, with only one sixth of such dry conditions accompanied by heatwaves; three observed ecological impacts characterized by vegetation damage and soil degradation; four underwent agricultural impacts manifesting as insurmountable obstacles in crop sowing and harvesting; and two reported hydrological impacts reflected in surface-water bodies. Moreover, drought impacts propagated into socioeconomic systems in four of the seven seasons and provoked wide-ranging ramifications in the sub-categories of food security, health, energy, and social stability, including, but not limited, to life-threatening food crises, lack of straw for fuel, epidemics along the railways, massive displacement and deaths, individual mental breakdowns, and the collapse of social ethics.

Twenty-four ways of responding were recorded in this event (Fig. 4d, Supplement D), three-quarters of which were adopted at the consumption level to alleviate socioeconomic impacts, particularly food insecurity. Despite little top-down direction for famine relief due to the power vacuum following the collapse of the Qing Dynasty, traditional measures, such as releasing local grain reserves, transporting grains from less-affected areas, and establishing soup kitchens in cities and towns, were adopted spontaneously by individuals, local officials, companies, and native and international relief societies. Meanwhile, a few actions were taken at the production and governance/management levels. Production-level responses include praying for rain, providing seeds to affected farmers, and digging new wells. While at governance/management level, efforts were put into maintaining social stability, such as resettling the displaced and reducing human trafficking. Compared to the aforementioned events, rebalancing goods supply and demand within drought-stricken areas with government involvement was

still identified as the main response pattern in this events, which also aimed to alleviate that huge pressures that drought placed on socioeconomic systems. Nevertheless, in the absence of a recognized central authority, self-help became a notable feature of this event. In contrast to the clear government-led mode in the Jing-Jin-Ji 1832 event, the involvement of government in this case dropped to 58.33%, while institution/organization and civil society were two emerging actors involved in 50% and nearly 30% of all documented responses, respectively. Meantime, individual/household participation here also increased by about 30 percentage points compared with the Jing-Jin-Ji 1832 event. In this context, a half of social responses included international disaster relief efforts.

**4.3 Prolonged drought with record-breaking heat in recent years**

The Germany 2018 drought event took place in an era where remarkable prosperity coexists with unprecedented challenges. On one hand, the country has developed into a more stable and well-functioning society with comprehensive modern economic sectors and, as a member state of the European Union (EU), deepened cross-boundary collaborations. On the other hand, ongoing climate change has been intensifying climate extremes, and the 2018 event, as a microcosm, has set a new benchmark for exceptional drought in Germany and even on a European scale (Conradt et al., 2023; Rakovec et al., 2022). Specifically, this event started in winter 2017/2018, reached its maximum in summer 2018 with as many as 33 specific effects and impacts of drought in 17 sub-categories, and lasted until spring 2021 (Fig. 3e, Supplement C). Thirteen of the 14 seasons experienced varying degrees of precipitation deficits that were accompanied by intense heatwaves in over one third of cases. Impacts belonging to the ecological, agricultural, and hydrological categories were reported in nearly three quarters, over five sixths, and about two thirds of the 14 seasons, respectively. In these three categories, broader repercussions of drought were observed when compared to previous events, as over three-quarters of drought effects and impacts in all nine sub-categories were documented. Furthermore, half of the seasons saw socioeconomic impacts, ranging from limited food price increases to short-term water conflicts; meanwhile, water scarcity became the most frequent emergency that society had to face.

Ninety-three ways of responding were taken during this event (Fig. 4e, Supplement D). Responses at the governance/management, consumption, and production levels accounted for 44.09%, 39.78%, and 16.13%, respectively. Governance/management became the level at which responses were most often taken, with particular attention paid to drought and/or heat impacts on ecosystems, hydrological systems, individual health, and economy. At this level, government was the most active participant in responses, and some efforts were put into developing long-term strategies and plans for disaster management and resilient development. For consumption-level responses, although their proportion dropped below 50% of all responses, they not only remained the most popular option when facing fodder shortages and water insecurity but also served as an important way to compensate for economic losses. As for responses at the production level, they were most concerned with agriculture (53.33%), especially crop performance. Farmers, as one of the most engaged actors in production-level responses, made efforts to minimize harvest losses in both behavioral and technological manners, including irrigating fields, postponing harvest dates, changing farming practices, and adjusting planting structures. In sum, the dashboard of this event (Fig. 4e) illustrated a response pattern that were characterized by broad concerns and multiple approaches, as drought effects and impacts in all five categories were dealt with, to some extent, by nearly a hundred actions. Compared with the 1834 and 1921 events in Germany, a further increase in the participation of government was noticed here, especially regarding financial and resource support from the Federal Government,

and the proportion of responses with government involved exceeded 50% for the first in this study area. Additionally, more and more responses were made in the technological and institutional manners towards better adaptation to climate change in the future. Nevertheless, "behavioral", "local", and "socioeconomic" remained the most often mentioned attributes of responses in this case, suggesting that alleviating exogenous pressures on the socioeconomic systems in drought-stricken areas was prioritized, both past and present.

The Jing-Jin-Ji Region also had an unforgettable experience with persistent hot and dry conditions in the late 1990s. At the time, improvements in technology and productivity since the 1950s had largely strengthened the coping capacity of the society, but growing scarcity of water resources increasingly constrained the economic development of this region (Zhang et al., 2009). The Jing-Jin-Ji 1997 event spanned 24 seasons from winter 1996/1997 to autumn 2002, culminated in summer 1997 with 16 specific effects and impacts of drought in 12 sub-categories, and peaked again in summer 2000 with 14 drought effects and impacts in 13 sub-categories (Fig. 3f, Supplement C). During this event, the lack of precipitation was observed in five sixths of all seasons, with intense heatwaves occurring in nine of the 20 abnormal dry seasons. Meanwhile, drought affecting ecosystems, agricultural production, and hydrological cycle were reported in half, seven-eighths, and five-eighths seasons, respectively. In the abovementioned categories, all sub-categories except for freshwater systems were noted to some extent, and unfavorable circumstances for soil moisture, livestock feeding, and groundwater level received attention for the first time in this study area. Additionally, 17 of the 24 seasons reported that drought disturbed the functioning of socioeconomic systems. By the end of this event, all six sub-categories belonging to the socioeconomic category were involved, and water scarcity replaced food insecurity as the most threatening issue.

Ninety-eight responses were documented in this event, with 34.69% at the governance/management level, 37.76% at the consumption level, and 27.55% at the production level (Fig. 4f, Supplement D). Consumption-level responses were the most employed, especially for ensuring food security, balancing water supply and demand, and alleviating the temporary financial stress on drought-affected farmers, companies, and local governments. Local authorities, sometimes under the guidance and financial support from the central authority, made most of these responses. Meantime, the private sector showed relatively high interests in improving water-saving technologies. Governance/management-level responses were designed to address the majority of documented drought impacts, with particular focuses on water scarcity and subsequent water conflicts. The central authority played an indispensable role in this case, especially with regard to technological and institutional responses. Production-level responses here showed the greatest concern for agricultural production (48.15%), aiming to reduce crop harvest losses, and attempts were made mainly in the manner of behavioral responses (e.g., irrigating fields) and technological responses (e.g., promoting dry farming techniques). In addition, plenty of efforts were also invested in the improvement of water facilities to reduce local water stress by increasing the availability of water resources. Similar to the Germany 2018 event, the response pattern in this event showed a greater richness of approaches and a broader range of concerns than historical drought cases, although the mitigation of socioeconomic issues within drought-stricken areas, especially those associated with supply-demand imbalances, remained the primary concern in recent drought cases. Meanwhile, increases were found in responses at the governance/management level and with long-term goals, indicating the growing recognition of the need for comprehensive disaster management and climate-resilient development. Furthermore, the top-down mode of disaster relief, i.e., local practices and cross-regional cooperation with guidance and support from the central authority, was again evident in this region at the

550 turn of the 21st century, which was obvious in the Jing-Jin-Ji 1832 drought but became less tangible in the chaotic early 1920s.

## 5 Discussion

### 5.1 Stable elements of drought impacts and response patterns

So far, according to textual information documented in Chinese, English, and German, this study has provided a
555 comprehensive and comparable profile of each selected extreme event in terms of categorized drought effects and impacts as well as different attributes of social responses. On this basis, a universalizing comparative analysis was first applied, which emphasized similar rules followed in different cases (Tilly, 1984), to reveal the stable elements of drought across regions and over time.

560 Out of the 17 sub-categories of drought impacts, seven can be recognized as universal key impacts in both study areas, as they were always mentioned despite the passing of time (Fig. 5a). These sub-categories are precipitation and temperature in the meteorological category, vegetation in the ecological category, crop in the agricultural category, river in the hydrological category, and food security and social stability in the socioeconomic category. Their recurrence across multiple drought cases revealed that anomalous weather, damage to vegetation,
565 unsatisfactory crop performance, insufficient river flow, hunger, and disorder were not only location-independent signals and/or threats of droughts but also inescapable and unresolved challenges in both historical and contemporary contexts. In addition, a number of other case studies on drought further confirmed the widespread nature of those key impacts, at least in other provinces of China (Chen et al., 2022), other European countries (Metzger and Jacob-Rousseau, 2020; Noone et al., 2017), south-eastern Africa (Klein et al., 2018; Pribyl et al.,
570 2019), the Middle East (Eklund et al., 2022), and North America (Burns et al., 2014; Hornbeck, 2023) during the 19th century to the early 21st century.

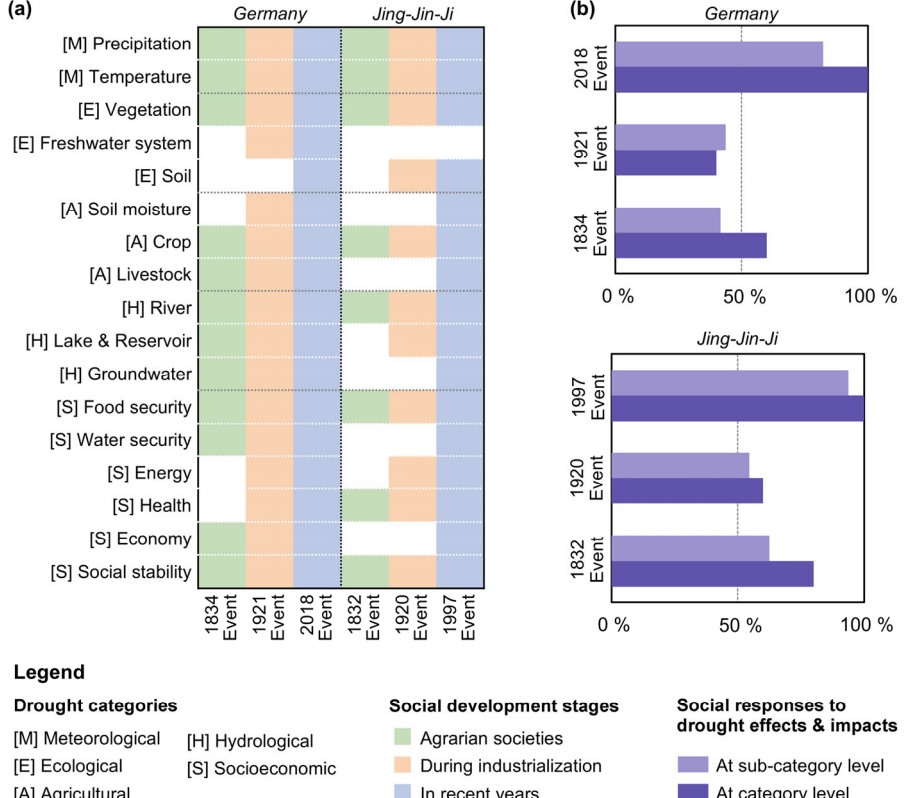

Figure 5 The documentation of and reaction to drought impacts in Germany and the Jing-Jin-Ji Region. The former is presented as (a) the categories and sub-categories of drought impacts that were mentioned in written documents at different social development stages. The latter is illustrated by (b) the percentage of drought categories or sub-categories to which society responded in all five categories or in all 17 sub-categories during each event

As for patterns of social responses, societies in the two study areas and at the three development stages shared a common preference for drought mitigation options, despite obvious differences in social-environmental contexts. It was found that societies tended to take reactive actions that intended to maintain or restore the supply-demand balance of scarce goods in drought-stricken areas, with the expectation of alleviating drought pressures on local socioeconomic systems and thus minimizing spillover effects. Governments were often actively involved in these responding processes, with or without a recognized central authority, indicating a consensus on the need for organization and coordination in the face of natural hazards (Fig. 4).

In other words, drought is drought. Regarding the pathways through which drought affects socioeconomic systems, it typically starts in the parts of human-natural systems that support basic human activities and then leads to imbalances in the supply and demand of life necessities in drought-stricken areas. As the organizational capability is gradually exhausted, the normal functioning of local societies is hindered, which, in some severe cases, causes individual deaths and social disruption and, ultimately, spills over to less-affected areas.

Nevertheless, positive signs for better coping with droughts have emerged in the move towards to a stable and modern society. The two study areas showed a similar trend in increasing efforts to reduce drought risk and damage through responses at the governance/management level and/or in an institutional manner (Fig. 4), which reflected growing attempts at the integration of short-term mitigation and long-term preparedness and rising interests in non-structural measures. Moreover, in both the Germany 2018 event and the Jing-Jin-Ji 1997 event, drought

impacts in all categories and over 80% of sub-categories received some degree of response (Fig. 5b), implying that societies became responsive to broader concerns beyond the basic needs of drought-stricken groups, although the effectiveness of these responses remains to be assessed.

## 5.2 Dynamic elements of socioeconomic impacts and coping strategies

To gain insight into drought-society interactions in different social-environmental contexts, comparison in this section was conducted with a special focus on the 24 specific drought impacts listed in the socioeconomic category (Table 1). As shown in Fig. 6, in the transformation from agrarian to modern societies, the diversification of drought impacts on socioeconomic systems can be observed in both study areas, reflected by increases in the proportion of documented impacts and in the number of involved sub-categories. Such changes can be attributed to two possible causes, namely increasingly complicated economic sectors and changing social concerns. The former has contributed to diversifying socioeconomic impacts, particularly in the sub-categories of energy and economy, since complicated economic activities usually encompass a greater number of intermediate links and a wider range of actors and thus lead to increased exposure to drought. The latter mays explain why water insecurity was only mentioned during the 1997 drought event in the Jing-Jin-Ji Region, when food insecurity was no longer the most pressing issue as famine had been eradicated, despite the fact that drinking water scarcity was an enduring problem in remote rural areas of this region (Wang, 2002).

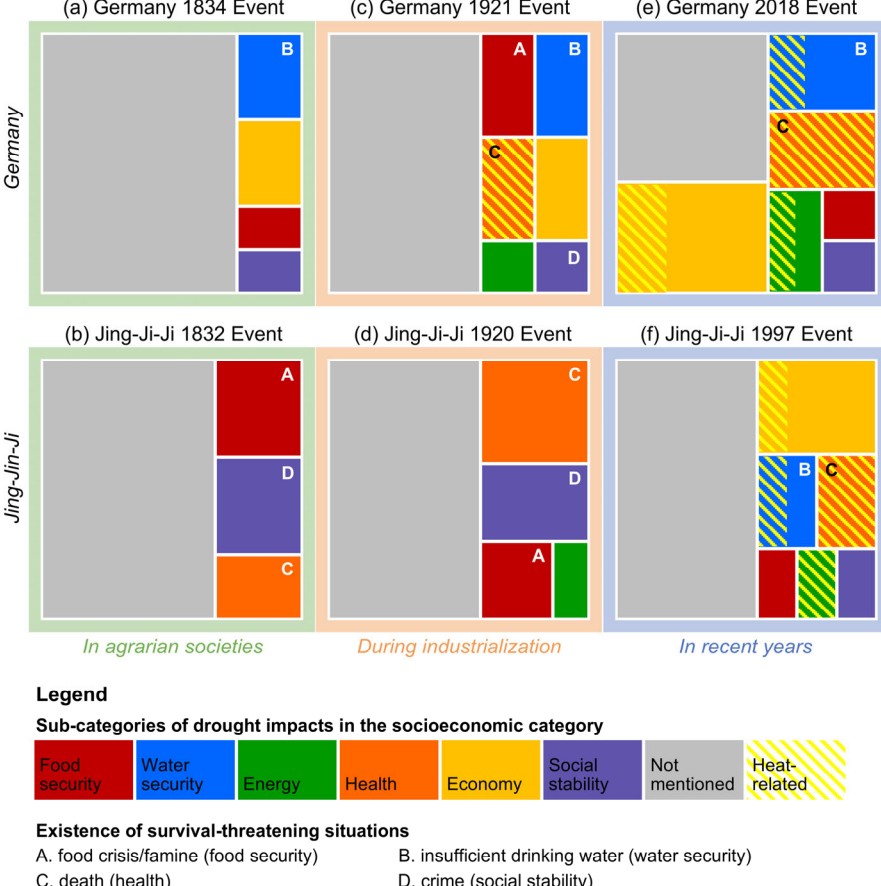

**Figure 6 Differences in drought impacts on socioeconomic systems in (a, c, e) Germany and (b, d, f) the Jing-Jin-Ji Region at different episodes of the transformation from agrarian to modern societies. For each event, the area of colored rectangles represents the proportion of impacts mentioned in each sub-category to all 24 impacts listed in the socioeconomic category. The area of gray rectangles denotes the proportion of all unmentioned impacts to the 24**

impacts in the socioeconomic category. The area of colored rectangles covered by yellow diagonal lines indicates the proportion of impacts related to heat in each sub-category. Heat-related impacts refer to those clearly described in written documents as being associated with or caused by abnormal hot conditions, such as increasing imbalances in water or electricity supply and demand due to increased consumption in hot weather, heat-related illness and death, and heat damage to infrastructure (e.g., railway, road)

Although drought impacts on socioeconomic systems have diversified in general, they are less survival threatening in the sub-categories of food security and social stability in the contemporary context (Fig. 6). This can be largely credited to the advanced responding toolbox in recent years, which allows for effective early and multiple interventions in the spread of drought impacts into socioeconomic systems. For Germany, an impact chain of precipitation deficit → harvest failure → rising food prices (→ crime) → food crisis was identified during the 1921 drought. At the time, the country was struggling with reparations, restrictions, and overall tough sociopolitical circumstances after the First World War. Under these circumstances, no response was recorded as taken prior to food security being affected, while the massive import of scarce products was not easy to realize. In addition, some actions against rising food prices, such as the decision to raise wages (Freiburger Zeitung, 1921a), actually pushed up food prices due to increased labor costs. As a result, the summer of 1921 witnessed an unmanageable food crisis caused by exorbitant prices of basic foodstuffs (e.g., bread prices rose by 40%), which was further exacerbated by criminal profiteering (Freiburger Zeitung, 1921b, c). While during the 2018 drought, the state of food security had been improved significantly, as evidenced by the controllable rise in food prices in the entire event. Such improvement was mainly achieved through early interventions in agricultural production (e.g., monitoring soil moisture, irrigating fields, and changing farming practices), multiple remedies to compensate harvest losses (e.g., expanding planting areas, postponing harvest date, and shifting to drought-resistant crops), and grain imports from France and Eastern Europe (Henning, 2018). Consequently, the development from precipitation deficits to food insecurity stopped before falling into survival-threatening food crisis, and crime, an intermediate link in the impact chain existing in the early 1920s, did not occur in this case.

As for the Jing-Ji-Ji Region, the impact chain of precipitation deficit → harvest failure → famine → crimes and deaths was a dominant pathway of drought causing mortality and disorder in agrarian societies and in the early years of industrialization. During the 1832 and 1920 droughts, famine-induced epidemics, displacement, violent crimes, and even massive deaths were frequently mentioned in spite of dozens responses to alleviate famine and its aftermaths (Li, 1993; Zhang, 2013). Different from the Germany 1921 event, unaffordable food prices were not seen as an indispensable link in the process from harvest failures to survival-threatening food insecurity (i.e., famine) in the Jing-Jin-Ji 1832 and 1920 events, which could be explained by the very limited degree of agricultural commercialization and population urbanization in this region (Wu, 1985). Nevertheless, societies, particularly those in volatile political circumstances, were prone to survival-threatening situations when food insecurity began to spiral out of control, regardless of whether top-down disaster relief or bottom-up self-help was implemented thereafter. When it came to the contemporary context, early interventions in agricultural production (e.g., irrigation, pest control, and late sowing techniques) and multiple remedies after harvest failures (e.g., expanding planting areas, replanting alternative crops, and shifting to drought-resistant species) helped reduce drought damage to crop performance during the Jing-Jin-Ji 1997 event, which was similar to the Germany 2018 event. Moreover, temporary shortfalls in food supply to some rural households (People's Daily, 2000a) were properly

resolved before anything got out of control, benefiting from the greatly improved national capacity to store and coordinate resources (e.g., grain reserve and allocation) in recent years.

In contrast to the successful breaking of pathways from precipitation deficits to food crisis/famine and subsequent social instability through the agricultural system, pathways from precipitation deficits to survival-threatening water insecurity (i.e., insufficient drinking water) via the hydrological system remained widespread in recent droughts, i.e., the Germany 2018 event (Fig. 6e) and the Jing-Jin-Ji 1997 event (Fig. 6f). To meet water demand during drought, various coping strategies were adopted in the two study areas, including, but not limited to releasing stored water, transporting water to affected areas, issuing water restrictions on the principle of ensuring drinking water, improving water facilities, and promoting water conservation. Simultaneously, actions related to sustainable water management were also taken in both study areas to mitigate present impacts and future risks of drought on the hydrological system such as insufficient river flows, inadequate reservoir storages, and low groundwater levels (Supplement D). In spite of these efforts, the lack of drinking water was still widely reported in the two events, as relevant responses were either difficult to maintain in a prolonged drought (e.g., storing water) or required a relatively long time to be effective (e.g., improving infrastructure). This suggests a greater need for anticipatory drought adaptation to ensure water security when compared to food security.

Last but not the least, extreme heat has become increasingly destructive in compound drought-heatwave events since the early 20th century and has gradually replaced precipitation deficits as the main climatic impact-driver of mortality in vulnerable population (e.g., infant and the elderly) (Fig. 6c, 6e, 6f). Furthermore, a wider range of heat-related impacts on water security, energy, and economy were observed in recent drought events (Fig. 6e, 6f), as compound drought-heatwave events became more frequent and intense in both Germany and the Jing-Jin-Ji Region (Fig. 3e, 3f). In addition to directly causing illness, death, and heat damage to infrastructure, heatwaves often exacerbated existing socioeconomic issues induced by precipitation deficits, such as increasing consumption of water and electricity and thus worsening supply-demand imbalances (Nicolai, 2018; People's Daily, 2000b). Unfortunately, although the two study areas could recognize and react to the additional effects of extreme heat within a short time, responses including issuing extreme weather warnings, giving health advice on heat and sun protection, and changing consumption habits and working hours were not able to eliminate those tangible and intangible impacts, even in modern societies. This indicates a common adaptation gap in the face of intensifying compound drought-heatwave events.

## 6 Conclusions

The general objective of this study was to explore the stable and dynamic elements of drought-society interactions in different socio-environmental contexts based on the comparable profiles of six representative drought events in Germany and the Jing-Jin-Ji Region (China) over the past two centuries. Adopting the area-specific reconstruction of dry-wet indices and multilingual written documents, this study first selected six extreme meteorological droughts in agrarian societies (i.e., Germany 1834 and Jing-Jin-Ji 1832 events), during the industrialization (i.e., Germany 1921 and Jing-Jin-Ji 1920 events), and in recent years (i.e., Germany 2018 and Jing-Jin-Ji 1997 events). Then, the progression of each selected event was established and analyzed according to categorized effects and impacts of drought and social responses patterns portrayed by five attributes, under the guidance of a common

impact-response structural framework. Finally, comparisons were conducted among the six events, focusing on the similarities and differences across regions and over time.

Seven key effects and impacts of drought and a major pattern of social responses were considered stable elements of drought-society interactions existing in different climate systems and sociocultural backgrounds. For key effects
and impacts of drought, abnormal dry and hot conditions, vegetation damage, unsatisfactory crop performance, insufficient river flow, food insecurity, and social instability were reported universally in both study areas and at all three social development stages, regardless of their severity, implying that they were non-negligible challenges to societies in both historical and contemporary contexts. As for the major response pattern, maintaining or restoring the supply-demand balance of scarce goods (e.g., food, water, and electricity) in drought-stricken areas
was a popular and practical option for different societies to mitigate drought impacts on the local socioeconomic systems and thus minimize spillover effects. Governments were commonly involved in such reactive actions, suggesting the need for organization and coordination in disaster management. Building on the above findings, an common pathway of drought impacts cascading into socioeconomic systems could be depicted: the impacts of abnormal weather conditions first appeared in the parts of human-natural systems that support basic human
activities, then caused supply-demand imbalance in life necessities at a local scale, next exhausted the organizational capability of drought-stricken societies, and eventually spilled over to less-affected areas when in situ responses failed.

Differences were visible in the specific implementations of disaster relief between the two study areas despite a
similar underlying logic (i.e., balancing goods supply and demand within drought-stricken areas), particularly with respect to the participation of central authorities. In the Jing-Jin-Ji Region, it was time-honored practice to tackle drought-induced supply-demand imbalances through local efforts and cross-regional cooperation—both official and non-official—under the direction and support of the central authority. Although relief efforts in the 1920 event were partially characterized by self-help due to a power vacuum after the collapse of the Qing Dynasty, cross-
regional and even nationwide allocation of resources and resettlement of displaced persons were still well documented, reflecting a tradition of mutual support arising from the long history of combating disasters as a unified country. While in Germany, the top-down mode where the central authority directs disaster relief and less-affected areas provide support was less pronounced. As time passed, national and subnational relief actions were increasingly mentioned in this study area, but their proportions in all social responses were consistently lower than
20% and 10%, respectively. In addition, the central authority here tended to promote disaster relief through less prescriptive approaches, such as negotiating with local authorities and/or non-governmental organizations (e.g., trade unions and associations) over financial aids.

It should be noted that the abovementioned differences in disaster relief modes were not directly responsible for
the varying socioeconomic impacts and response effectiveness among events. In contrast, changes in drought-society interactions were often observed when comparing drought events at different social development stages, rather than in different environmental and cultural backgrounds. As moving towards modern societies, both Germany and the Jing-Jin-Ji Region witnessed increases in the amount and type of documented socioeconomic impacts, which could be explained by increased drought exposure due to more complicated economic sectors and
broader social concerns beyond traditional drought challenges. Meanwhile, drought became less threatening to

individual survival in the contemporary context, especially with regard to food security. Prior to modern societies, despite considerable efforts to reduce food shortages, the two study areas were generally prone to survival-threatening food insecurity (i.e., food crisis or famine) during extreme droughts, particularly in the volatile political climate of the early 1920s. Food crisis and its aftermaths (e.g., crimes and deaths) were mentioned from time to time, whether or not disaster relief was implemented in a top-down mode. While in recent years, societies in both study areas were able to take early interventions in agricultural productions and multiple remedies after harvest failures, which cushioned the impact of drought on crop performance, and resolve temporary shortfalls in food supply before things got out of control through either efficient domestic resource allocation or extensive international food trade.

Nevertheless, compared with the remarkable progress made in ensuring food security, survival-threatening situations in the sub-categories of water security (i.e., drinking water scarcity) and health (i.e., heat-induced mortality) still existed in both the Germany 2018 event and the Jing-Jin-Ji 1997 event. An underlying cause was that the potential effective responses to these situations (e.g., improving infrastructure, adopting sustainable water management, and building climate-resilient city) could not be expected to take effect as immediately as actions to address food insecurity. This revealed a common adaptation gap in mitigating climate impacts on water security and health and urged anticipatory adaptation in the face of intensifying compound drought-heatwave events.

**Data availability**

All the data used in this study are available in Supplement C and Supplement D.

**Author contribution**

DZ and RG contributed to the development of research concept and methodology. RG and MK performed the collection and extraction of German documentary data. DZ collected Chinese documentary data, analyzed textual information from both study areas, and prepared the manuscript with contributions from all co-authors. All authors have read and commented on the latest version of the manuscript.

**Competing interests**

The authors declare that they have no conflict of interest.

**Acknowledgement**

The authors would like to thank Siyu Chen for her help in collecting historical disaster data in China, Dr. Nils Riach for his advice in improving article structure, and the China Scholarship Council (grant no. 202106040016) for the financial support.

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
