# Peer review of "Stable yet dynamic: A cross-era comparative case study of drought impacts and social responses in Germany and Jing-Jin-Ji Region (China)"

_EGUsphere, 2024_

## Author Response (AR1)

**Reply to comments from Reviewer #1**

We kindly thank the reviewer for providing such thoughtful advice and comments, which are particularly helpful in clarifying methodological choices, better interpreting results, and reducing language ambiguities in this manuscript. Please see below for point-by-point responses. In the reply, comments are quoted in *italics*, and our responses are interleaved in regular font. In addition, line numbers, sections, figures, tables or supplementary materials indicated herein refer to the revised version of manuscript, and they are marked green when pointing out the location of specific modifications in the revised manuscript.

*This study undertakes a valuable comparison of historical drought events and impacts in a region of China and in Germany. It introduces an elaborate and fine-grained classification system to analyze drought impacts. The choice of topic and case studies is promising, and the data collection and analysis performed for this study is valuable. Nevertheless, the manuscript should undergo revisions and further review before publication.*

**Response**: Thank you very much for the positive comments on our study, as well as for pointing out where we can improve this manuscript.

*First, the language requires some editing and correction before further review. In some places, the meaning of the text is unclear. For example, lines 22-23 refer to "a common responding preference of societies under different circumstances." This does not literally make sense. I think it means "a common and preferred societal response across different circumstances." However, it might also mean something like "a common preference of societies responding to other (hazardous) circumstances." If this were only an issue of style, then I might leave it to text editors later in the publication stage. However, the lack of clarity has made it difficult to properly evaluate some methods and claims of the study. Therefore, it should be addressed before further review.*

**Response**: Sorry for the unclear text. "A common and preferred societal response across different circumstances" is what we meant. To avoid misunderstanding, we have amended the unclear expression "a common responding preference of societies under different circumstances" to "Despite distinct socio-environmental contexts and different disaster relief modes (e.g., top-down or bottom-up), maintaining or restoring goods supply-demand balance was an underlying logic of drought mitigation shared by different societies" (Lines 20-23).

Accordingly, we also checked the other parts of our manuscript and modified similar unclear expressions as below:
Section 5.1, Line 554:
"5.1 The unchanging nature of drought impacts and responding preference" has been amended to "5.1 Stable elements of drought impacts and response patterns".

Section 5, Lines 580-581:
"In terms of social responses to drought, a common preference for coping strategy…" has been amended to "As for patterns of social responses, societies in the two study areas and at the three development stages shared a common preference for drought mitigation options, despite obvious differences in social-environmental contexts."

"seven key drought impacts and a common responding preference were observed…" has been amended to "Seven key effects and impacts of drought and a major pattern of social responses were considered stable elements of drought-society interactions existing in different climate systems and sociocultural backgrounds."

*A second and related issue regards the ambiguity in some key terms. "Drought" is used in the manuscript in different ways, including (1) precipitation deficit, (2) unavailability of water for agriculture, and (3) drought as a disaster (i.e., a precipitation deficit that has caused negative societal impacts). This ambiguity is especially a problem in the introduction section, before the authors have defined their terms and their framework of analysis.*

**Response**: We are sorry for the ambiguity in term drought, but we would like to explain why a universal definition of drought was not given in the introduction section.

To reflect differences in regions, needs, and disciplinary approaches, drought should be–and often is–defined according to specific application. Thus, a precise and universal definition of drought is still absent and might be of little value (National Drought Mitigation Center, 2025; Wilhite et al., 2014). For this reason, it is difficult to bring all the literature we reviewed in the introduction section under a particular drought definition, because, in many cases, drought was generally treated as an extreme event or shock with a variety of natural and social consequences, especially when research focused on exploring drought-society interactions. In sight of this, we would suggest considering drought, especially in the introduction section, as a climate extreme that leads to water shortages and corresponding consequences in different components of the socio-ecological systems.

Nonetheless, we still attempted to reduce the ambiguity in term by using specific phrase (e.g., precipitation deficits, insufficient rainfall) when emphasizing the meteorological aspect of drought. For example, "…individuals or groups in different socio-environmental contexts when facing the same climatic stimulus (i.e., precipitation deficits)…" (Lines 66-67) in the Introduction section. Additionally, in order to reduce confusion, modifications have been made to the Introduction section (Lines 36-37) where the term drought first appears as follows: "Drought, as a climate extreme that can lead to water shortages, is thought to have the largest adverse impacts of all weather-related natural hazards throughout human history…"

In fact, the ambiguity in the term "drought" is also a reflection of the complex nature of drought, which challenges the comprehensive characterization and comparison of different drought cases. This was an important motivation for us to develop the drought categories (i.e., the sub-framework I of figure 2 and Table 1) before undertaking comparative studies.

**Reference**

National Drought Mitigation Center: Types of Drought, https://drought.unl.edu/Education/DroughtIn-depth/TypesofDrought.aspx (last access: 12 March 2025), 2025.

Wilhite, D. A., Sivakumar, M. V. K., and Pulwarty, R.: Managing drought risk in a changing climate: The role of national drought policy, Weather and Climate Extremes, 3, 4–13, https://doi.org/10.1016/j.wace.2014.01.002, 2014.

*It is also a problem in sub-framework I of figure 2, which includes "precipitation deficit," "heatwave," and "soil moisture" as "impact categories." Normally, these items would be potential definitions of drought (i.e., meteorological drought, hydrological drought, etc.). Including these as "impacts" of drought raises the question of how the drought itself is defined.*

**Response**: Many thanks for the reminder. We agree that it might be problematic to include these items as "impact categories." We think the problem lies in the mismatch between the words/expressions we use and the meaning we intend when naming and explaining the sub-framework I of figure 2. This problem also exists in Table 1, as you mentioned in your following comment.

Our intention was to develop a framework that involves drought definitions from different perspectives (i.e., meteorological drought, hydrological drought, etc.), or rather, covers different types of water shortages/imbalances occurred in different components of the socio-ecological systems (e.g., ecosystems, socioeconomic systems). After careful consideration, we have decided to label the sub-framework I simply as "drought categories," which is probably more straightforward and the closest to our intention. Corresponding modifications will also be made to Figure 2, Figure 5, Table 1, and the main text (Lines 15, 80, and 224).

*Table 1 also lists a series of "manifestations" of drought; and this, too, seems ambiguous. Do "manifestations" refer to instances of drought impacts that we can infer from the historical evidence? Or do "manifestations" refer to the ways that observers observed and recorded the occurrence of drought? Or both of these? For example, is a heatwave a "manifestation" when (a) we have instrumental or phenological records indicating extreme heat, (b) historical sources complain that heat was affecting health or crops, or (c) both? The difference is subtle but important. Are we trying to use historical evidence in this case to (objectively) reconstruct impacts, or to understand how those impacts were perceived and experienced at the time, or both? Moreover, we must be cautious whether the available evidence is more likely to record certain types of impacts than others.*

**Response**: Sorry for the inappropriate word choice. We think that "effects and impacts" might more aptly refer to what we have listed in each drought category, as this phrase encompasses both the outcomes of a phenomenon (e.g., precipitation deficits) and how they affect something else (e.g., drought-induced harvest failures). Accordingly, we have replaced "manifestations" with "effects and impacts" or other more appropriate expression depending on specific contexts in Table 1, Figure 6, Supplement C, and the main text (Lines 24, 29, 225, 229, 309, 346, 386, 417, 449, 484, 489, 497, 520, 521, 602, 604, 607, 609, 627, 736, 739-740, 753, and 755).

In addition, we would like to explain a little bit more about the sources we adopted to establish the progression of drought events, as well as how they were used. As we explained in the section 3.1, the establishment of drought progression relied on textual information derived from written documents (i.e., drought impacts that were documented) rather than instrumental records. This could be seen as a trade-off for the situation where instrumental measurements were less available to historical cases but drought-related descriptions were common in both the past and present, which also reduces, to some extent, the uncertainty stemming from different ways of observation. For example, both the summer of 1834 and the summer of 1921 could be considered too hot, according to the descriptions of "der Sommer äuserst trocken und heiss (the summer is extremely dry and hot)" in 1834 and "der

außergewöhnlichen Hitze (the exceptional heat)" in 1921, respectively. Although there were several temperature records for the summer of 1921 (e.g., 39.4 °C in Karlsruhe on 28 July 1921), they served more as support for the abovementioned textual information than as specific criteria for identifying heatwaves alone.

*Some of the attributes of societal responses to drought (sub-framework II) are also unclear. For example, under "dimension," production-level responses are defined in terms of minimizing drought impacts on producing primary products. Yet consumption- level responses also include "expanding supply." How do we distinguish in practice between the minimizing loss of production and attempting to expand production? Specific examples would be helpful.*

**Response**: Many thanks for the advice. Production-level responses focus on the process of creating outputs, such as cultivating crops, raising livestock, obtaining water from natural resources, etc. Taking grains as an example, both direct reduction of drought-induced yield losses (e.g., irrigation) and expansion of cultivation areas to compensate for yield losses belong to taking remedial actions to minimize drought impacts on the production of grains. While consumption-level responses pay more attention to supply and demand and have less to do with the production process. Again, taking grains as an example, "expanding supply" here refers to increasing grain supply in drought-stricken areas by transporting from less affected areas or releasing reserves to the market, rather than putting effort into improving crop yields or expanding cultivation areas. Several specific examples have been added to these two levels of responses (please see below) to help with understanding.

Section 3.3, Lines 245-248:
"Production-level responses intend to minimize drought impacts on producing primary products by taking remedial actions based on current resources (e.g., irrigation, replanting short-season crops to compensate for harvest failures) or exploiting environmental potentials (e.g., digging wells deeper)."

Section 3.3, Lines 248-253:
 "Consumption-level responses aim to balance the supply and demand of a particular good or service by expanding supply (e.g., transporting food or water to drought-stricken areas, releasing grain reserves to the market), reducing waste and/or outflow (e.g., promoting water conservation), increasing purchasing power (e.g., increasing remuneration), restraining demands (e.g., water restrictions and  rationing), adjusting allocation strategy (e.g., prioritizing domestic consumption in case of electricity shortages), and finding substitutes (e.g., eating famine food such as ground corn cobs and tree leaves)."

*A third issue relates to the study's method of counting impacts and responses within a fine-grained analytical framework. Long-term historical studies of climate or weather impacts all face a similar challenge: namely, that it can be difficult to distinguish real long-term changes in impacts from changes in the quantity and nature of historical evidence about those impacts. Normally, studies attempt to minimize this problem by utilizing relatively complete and consistent historical records. To some extent, that is the case in this study. However, by analyzing the evidence for so many possible categories of impacts (graded by dimension, type, scale, actor, and target), the study exacerbates the risk that even small changes in the availability and nature of the evidence will influence the distribution and frequency of events in each category. While this problem does not invalidate the study's*

*framework, it does require caution in the interpretation of results. Moreover, it suggests that the study should aim to do more than simply assess the distribution of impacts in each category and try to derive conclusions from those distributions. The study would have been much stronger if it had started with theories about drought based on past studies and then aimed to test these theories utilizing this new comparative method and analytical framework.*

**Response**: We fully agree that changing quantity and nature of historical evidence is a common challenge in documentary-based historical studies of climate or weather impacts, especially those covering long time spans or involving multiple episodes of different eras. This challenge was also a thing that we had to think about at the early stage of our study. On one hand, it was inevitable to adopt data from multiple sources and in different forms when conducting comparisons across regions/countries and periods. On the other hand, some degree of quantification was needed in the face of hundreds of impacts and responses in the six selected drought events; otherwise, the manuscript would be particularly lengthy due to the itemization of historical evidence.

Therefore, we used the strategy of categorization instead of simply counting the mentions of a given phenomenon in written documents, to reduce the influence/risk of changes in the quantity and nature of historical evidence on the analysis of impacts. In other words, impacts extracted from various historical evidence were first categorized into corresponding classes in the five drought categories (i.e., meteorological, ecological, agricultural, hydrological, and socioeconomic categories, please see Fig.2 and Table 1 for details), and then quantifications and comparisons were conducted based on those categorized impacts. For example, if there were two mentions of the lack of drinking water during a dry summer in a historical event and ten similar reports in a modern event, we would considered it as one occurrence of water insecurity in a particular summer in both historical and modern events, rather than water insecurity occurring twice in the historical event and ten times in the modern event. With such processing, changes in the number of records would not be captured as changes in the effects of drought on water security when comparing the historical and modern events. We hope this example helps to demonstrate how this strategy can narrow the gap between past and present documentation in terms of record abundance and information density.

Of course, we are not expecting that our strategy is able to eliminate all the possible influences (or rather uncertainties) of changes in historical evidences about impacts with time, which is the reason for using phrases such as "not mentioned/documented" rather than "not existing" when interpreting our results. However, we would still like to argue that this strategy mitigated rather than exacerbated "the risk that even small changes in the availability and nature of the evidence will influence the distribution and frequency of events in each category" in this study. Let us again use the example of water security. In a given event, the occurrence of water insecurity should be identified based on the totality of available evidence about the lack of water for drinking or other usage. That is, the number and type of seasons in which drought affected water security would only be influenced when evidence about drought-induced water insecurity during this event was missing entirely from the multiple sources of written documents we adopted (supplement A) or completely unavailable at the seasonal scale.

As for response, a similar but slightly different strategy was employed to process their historical evidence and then portray them from five attributes (i.e., dimension, type, scale, actor, and target). Please see our reply to the next comment for more information on the processing of historical evidence about response.

Moreover, it is an important aim of case studies to first create a comprehensive profile of each case and then derive conclusions from these profiles. Regarding our event-based comparative study, it is fundamental to providing the portrayal of selected droughts in a comparable manner, as it demonstrates how the method and framework we proposed can support documentary-based comparisons across regions (languages) and over time. Meanwhile, uncovering similar and different drought-society interactions in distinct socio-environmental contexts is a major concern of our case comparisons, especially disclosing the elements that have or have not changed with social development, because this can provide historical perspectives and valuable references for understanding and addressing drought challenges at present and in the future. Nevertheless, we are very happy to know that our new comparative method and analytical framework have the potential to contribute to drought theories (e.g., testing some of them). Many thanks for this kind remark. Although our current attempts have not really been able to involve theories of drought, we are willing to make further efforts in this direction as more drought cases are collected and analyzed.

*Starting at line 295, the analysis in the study focuses largely on quantifying the ratio of responses in different categories and at different levels. However, it is not clear how to interpret these ratios—or indeed, whether the relative quantities of responses in different categories are historically meaningful. For example, let us take the following statement on lines 297–299: "The local government was the most engaged actor who dominated 50% of responses at this level, while the individual/household and civil society also took a few actions to prevent livestock starvation and secure community's drinking water supply, separately." To begin with, I was not sure how to interpret this statement. Do the authors mean (1) that they reviewed the historical evidence as a whole and interpreted it to mean that local government took the leading role in drought response, and moreover that this interpretation is reflected in the fact that 50% of drought responses involved local governments? Or does it mean (2) that the authors categorized all references to drought responses and that 50% involved local governments, and thus from this ratio they have inferred that local governments must have taken a lead in responding to drought? If they meant (1), then the 50% figure is more illustrative than conclusive. If they meant (2), then there could be problems with inferring the significance of the local government response just from the number of mentions in the sources. After all, there are more local governments than central governments, so they have more responses, even though those local responses might have been smaller than responses directed by central governments. Local governments also produce more records than households or non-government agencies, meaning they may appear more in historical records, and so on. In short, I felt the study needed to demonstrate why and how these numbers were meaningful— and not just artefacts of the source availability or the methods of analyzing the evidence.*

**Response**: We are sorry for not explaining our purpose and strategy of quantifying responses clearly, as well as for the ambiguity in interpreting the results of response analysis. Here, we would like to clarify two points:

i. **Quantification strategy of responses and meanings of ratios/numbers**

Similar to the processing of evidence about impacts, we reviewed the historical evidence associated with a particular event as a whole and identified how many different responses were implemented after integrating different records of the same action into one way of responding, as opposite to simply count the times a specific action was mentioned during this event. For example, in a given event, we would only consider irrigation as

one way of responding, not three, even if there were three mentions of farmers (same or different persons) irrigating their fields (at the same or different times or places) due to rather dry conditions.

Different from the quantification of impacts, instead of classifying responses into different categories and calculating the occurrences of each category, we first labeled each response with one or more classes from each of the five attributes (i.e., dimension, type, scale, actor, and target), then grouped all responses during a particular event according to their dimensions (i.e., at the levels of production, consumption, or governance/management), and finally calculate the proportion of different classes in the four other attributes within each group. In other word, responses were depicted one by one from five common aspects, as opposite to responses falling into five different categories, and the results of quantification (i.e., ratios/numbers of response attributes) were supposed to help illustrate, or rather visualize, the patterns of responses to drought in different regions and at different episodes of social transformation in a comparable and relatively holistic manner.

In addition, we would suggest viewing the per-event quantification results of responses shown in Fig. 4 as response pattern dashboards outlining levels, manners, spatial extents, social involvement, and key concerns of drought coping strategies in different socio-environmental contexts. To aid understanding, we have changed the caption for Figure 4 to "Dashboards of social response patterns in selected drought events, outlining levels (Dimension), manners (Type), spatial extents (Scale), social involvement (Actor), and key concerns (Target) of drought coping strategies in different regions and at different social development stages…"

ii. **Confusing statement on lines 297–299 in the original version of manuscript**
Interpretation (1) mention in this comment is closer to what we intended to mean, i.e., local government was visible as one of the actors in half of the consumption-level responses during the Germany 1834 drought event. Namely, 50% figure is illustrative of governments' active participation in maintaining or restoring the balance between supply and demand, rather than inferring the undisputed leading role or local governments or the significance of the local government response.

We think the problem with understanding this statement is probably caused by the inappropriate word choice. The word "dominated" is indeed misleading. Meanwhile, the use of "local" and "central" as modifiers before "government" might hinder comprehension as well, since "local" also refers to one of the classes of the response attribute *Scale*. In fact, both local and central government belonged to the class *Government* of the response attribute *Actor*, and they were not calculated separately when analyzing response patterns. Our intention in adding these modifiers was to provide some details on this actor in the main text. To avoid confusion, we use the phrase "central authority" in the revised manuscript when emphasizing the involvement of national governments (e.g., the Federal Government in Germany) (Lines 269, 366, 403, 409, 471, 535, 538, 550, 585, 722, 724, 727, and 731).

In short, we have fully recognized the risks of drawing conclusions about historical climate-society interactions based simply on the number of mentions in the sources, especially after reading your constructive comments, and we have made a lot attempts at mitigating such risks in the current study. However, we did not explain these

attempts clearly enough, which hindered the understanding of our methods, and our wording in interpreting results was somewhat imprecise and even misleading. Accordingly, we have reorganized the section 3.4 (Lines 279-357) to explain quantification strategies in a clearer way and amended ambiguous wording in interpreting of our results in the section 4 (Lines 360-373, 396-411, 460-476, 494-514, and 531-552). Please see the relevant text in revised manuscript for details on modifications.

*Fourth, the study should acknowledge certain limits regarding its selection of events for analysis. This is a study about three of the worst meteorological droughts in the two target regions. It is not a story about drought disasters and/or resilience overall across the study period. There were, presumably, other periods of droughts during these centuries that had greater or lesser societal impacts, depending on historical conditions, political decisions, and cultural responses. Had the study selected three of the most impactful droughts (in terms of societal consequences) rather than three of the strongest meteorological droughts (as measured by the dryness-wetness indices) then it may have revealed different patterns of vulnerabilities and societal responses. Similarly, the study could have contrasted examples of droughts with greater societal impacts and droughts with lesser societal impacts in order to identify differences. All of this is not to say that the authors' choice of case studies was incorrect or unhelpful. Nevertheless, the authors should explain what this selection of events for analysis can or cannot reveal compared to other potential selections.*

**Response**: We think there might be some misunderstanding here, which is probably caused by imprecise expressions such as "drought-society interactions during the transformation from agrarian to modern societies (in abstract)." As emphasized in the title of this manuscript, we carried out case studies based on empirical evidence of drought-society interactions in specific matrices of societies. In other words, we intended to give a comprehensive and comparable profile of each selected drought event through an in-depth examination of a specific unit, which served as the basis for identifying similar and different drought-society interactions in distinct natural and social contexts, rather than to establish a continuous drought storyline from the 19th century to the present so as to provide a full picture of drought disasters and/or resilience over the past 200 years. In this case, "drought-society interactions at different episodes of the transformation from agrarian to modern societies" could be closer to what we intended to mean. We have checked through the manuscript and modified the expressions that might cause such a misunderstanding.

Nevertheless, we agree that different selections of events for analysis might reveal different patterns of vulnerabilities and societal responses, as the development from a hazard to a disaster is often influenced by or even dependent on how society responds based on its resources and capabilities at the moment. This was also our motivation for conducting event-based comparative study involving different stages of social development, rather than long-term time series analysis.

We selected the strongest meteorological drought with the aim of capturing the performance of natural and social systems under the greatest exogenous pressures from precipitation deficits in specific socio-environmental contexts. Feasibility was also one of our main considerations. It is difficult to determine a standard for identifying the most impactful droughts in terms of societal consequences, especially when involving different sociocultural contexts and changing social morality. For example, should the most impactful droughts be the ones that caused

the biggest obstacles in daily life, the largest number of deaths, the greatest economic losses or the most dramatic social disturbances (e.g., fall of the regime or even societal collapse)? In addition, societal consequences should be part of the outcomes of drought-society interactions, namely part of what we were trying to reconstruct. It smacks of reversing the research process if events were first selected in terms of societal impacts then reconstructed and analyzed.

Overall, we would like to further explain our motivations and objectives for conducting case studies and selecting events based on meteorological extremity, which might not be adequately explained in the text. Thank you very much for the constructive comment. Accordingly, we have mainly modified the Section 3.2 as below. We hope that this would enhance the interpretation of our event selections.

Section 3.2, Lines 167-176:
"The selection of drought events was based on the extremity of dry conditions and the representativeness of different social development stages. In other words, for a selected event, its precipitation deficits should be severe enough to be considered as one of the greatest exogenous pressures on a specific region with respect to climatic conditions; meanwhile, its occurrence should be at a certain episode in the profound transformation from agrarian to modern societies that human society experienced over the last 200 years (IPCC, 2022). Following this principle, it is expected to capture the performance of socio-ecological systems under the pressure form extremely dry conditions in representative socio-environmental contexts, although the drought events ultimately selected are not necessarily the most socially disruptive ones."

*The discussion about crises on lines 179-180 seems misleading, since that is not really the focus on the study. Additionally, as the authors note: "This study took the onset and cessation of precipitation deficits as the beginning and end of an event." Although this certainly simplifies the analysis, it means that the study will not capture some cascading effects of droughts that take more time to develop and to appear in historical records. Moreover, it also makes it harder to detect proactive measures to reduce impacts before droughts occurred. These limitations should be acknowledged when discussing the selection of episodes for analysis.*

**Response**: Many thanks for pointing out the misleading. We have reconsidered the content about crises in the Section 3.2 and changed it to expression as follows. We think the new expression matches better with our considerations of event selection mentioned in our response to the last comment.

Section 3.2, Lines 192-193:
"In other words, based on currently available data, it is impracticable to ensure that the two study areas were always under the same strong exogenous pressures from precipitation deficits in selected drought events. Thus, this study turned its attention to the most significant deviations (i.e., the driest years) reflected in the dry-wet indices applied to each study area and then set criteria independent of the location and data."

As for taking "the onset and cessation of precipitation deficits as the beginning and end of an event," we believe that this is a necessary simplification regarding different data availability and social concerns between the past and present, as exemplified by increasing focuses, improving monitoring techniques, and abundant reports on soil

moisture (e.g., varying degrees at difference soil depths) nowadays but very limited (e.g., mentioned only in severe cases like "cracks in land") or even no relevant records in history. It could be seen as the greatest common denominator of drought narratives across regions and over time that the onset and end of precipitation deficits were always well documented with relatively clear temporal information, regardless of severity. In addition, unlike harvest failures, reports of precipitation deficits were not limited to specific seasons tied to plant life cycles.

Certainly, we acknowledge that it is not possible to avoid completely the uncertainties arising from under-recording, just as historical studies of climate or weather impacts generally face. However, we think these uncertainties would not invalidate the main findings of this study. For the two sources of uncertainties mentioned in this comments alone:

i. **About capturing some cascading effects of drought that take longer time to develop**

It is true that the development of different types of drought effects and consequences can take from several weeks (e.g., 1–2 months dry conditions can cause failure in rain-sensitive horticultural products and hay) to more than ten months (e.g., water shortage from a year can slow down production economy) (Glaser and Kahle, 2020, Table 3). In particular, it often takes longer for drought impacts to reach and cascade through socioeconomic systems in a tangible manner when compared to ecosystems, agricultural systems, and hydrological systems, since the development of socioeconomic impacts largely depends on the capability of a society to cope with drought.

Nonetheless, it was still possible (and has been successful) to depict the prolonged and complicated impact chains from abnormal weather conditions to high-level socioeconomic consequences (e.g., individual deaths and social instability) in our event-based study focusing on the duration between the start and end of precipitation deficits. One reason could be that our selected extreme events were relatively long-lasting, spanning from five seasons to six years, which, in the shortest case, offered a time frame longer than one year for the emergence of diverse drought effects and consequences. Another reason could be the general sensitivity and concern of societies about drought, the climate extreme with which humans have long struggled, which facilitated the relatively timely and detailed documentation of drought consequences, especially those closely related to livelihoods (e.g., farming).

To make a rough assessment of potential omissions of drought impacts, we took another look at drought-related descriptions in chronicle and compilation, yearbook, and secondary research, i.e., written documents that were normally composed in the form of after-the-fact summaries. It was found that tangible impacts reported after sufficient rainfall were more or less a continuation or repetition of previously reported impacts rather than entirely new ones, except for "the harvests of 1922 and 1923 were good, but it was hard to hire agricultural workers because so many had gone to the cities (Li, 2007)" (i.e., potential for affecting crop harvests) shortly after the Jing-Jin-Ji 1920 drought. The influence of such omission on categorization-based analyses of drought cascading and its socioeconomic consequences seemed to be acceptable. While for less tangible but far-reaching socioeconomic impacts, such as altering resilience and even accelerating social collapse, they are important but somewhat beyond the focus of our event-based case study, i.e., to provide and compare profiles of drought-society interactions in different regions at different episodes/cross-sections of social transformation. Long-term time series analyses would be more helpful in terms of this topic.

**ii. About detecting proactive measures to reduce impacts before droughts occurred**

We think that "proactive measures to reduce impacts before droughts occurred" could be considered in two ways:

(a) Taking short-term actions to mitigate the risk/possible loss of a certain impact, for example, storing water in advance in case of future water shortages due to drought development. This is indeed a proactive measure to prevent drought from affecting water security, but actually, it is also a reaction to observed and/or monitored abnormal conditions such as the lack of rainfall for some time. In this situation, it is possible to detect such proactive measures within the event duration we defined, as these measures were implemented after the first occurrence of precipitation deficits.

(b) Reducing vulnerability to drought through long-term resilience building, such as improving water management, upgrading infrastructure, planting drought-resistant trees and crops, establishing anti-drought organizations, etc. This was relatively seldom seen in records of past drought events but was often mentioned during periods of precipitation deficits in recent events (i.e., the Germany 2018 and Jing-Jin-Ji 1997 events). Conservatively speaking, such proactive measures have been encompassed to some extent in our current study, although it remains hard to attribute precisely the absence of these measures from historical drought cases to omissions or a lack of existence/documentation. As for measures taken prior to the start of our selected events, we considered it more likely to be a lesson learned from previous droughts than a result of the target events.

Nevertheless, after working on this comment, we have realized the importance of clarifying the limitations in episode selections and the inadequate explanation in the original manuscript. Thus, modification has been made to the section 3 as follows. Thank you very much for the advice.

Section 3.4, Lines 279-289:

"Providing comprehensive and comparative profiles of selected drought events (i.e., drought progression) was the first step in comparing the drought-society interactions they implied. This study took the onset and cessation of precipitation deficits as the beginning and end of an event, which was a trade-off between the limited availability and relatively narrow scope of drought records in history and the abundant and comprehensive drought reports at present. As sequels to the lack of precipitation, other effects and impacts of drought may not take place (e.g., electricity shortage) or be well documented (e.g., degree of soil moisture at different soil depths) in every region and at every episode of social development. In addition, some impacts can continue in an inconspicuous manner over a non-measurable span after precipitation returned to normal (e.g., reduced coping capacity for future hazards), which goes more or less beyond the scope of the current case study. Nevertheless, it should be noticed that this definition of event duration is not ideal for exploring drought effects in the long run (e.g., post-drought labor shortages due to climate exodus) or drawing a complete picture of drought resilience over a period."

**Reference**

Glaser, R. and Kahle, M.: Reconstructions of droughts in Germany since 1500 – combining hermeneutic information and instrumental records in historical and modern perspectives, Clim. Past, 16, 1207–1222, https://doi.org/10.5194/cp-16-1207-2020, 2020.

Li, L. M.: Fighting famine in North China: State, market, and environmental decline, 1690s~1990s, Standford University Press, Stanford, 2007.

*Since the Chinese droughts here have been defined here only in terms of dryness/wetness indices, it may be helpful to note other measurements of drought during the years under examination, such as tree-ring based reconstructions. In any case, it would help if the reader were better able to compare the severity of drought as measured in the dryness/ wetness indices with the severity of drought as measure by SPEI.*

**Response**: Indeed, the dryness/wetness indices might not be the ideal data for measuring the magnitude of meteorological drought in a very precise, quantitative, and comparable manner; but unfortunately, it is the best available to us at the moment in terms of resolution and spatiotemporal coverage.

As you suggested, we have checked International Tree-Ring Data Bank (ITRDB), the world's largest public archive of tree ring data with more than 5,000 sites, for possible tree ring data applicable to our study area in China. However, tree ring data are not available for our study area or even neighboring areas (Fig. R1). We have also thought about the Old World Drought Atlas by Cook et al. (2015), but it can only cover our study area in Europe (i.e., Germany), thus is inapplicable to our study area in China (i.e., the Jing-Jin-Ji Region).

[Figure]

Figure R1 Locations of tree ring data in the ITRDB (https://www.ncei.noaa.gov/products/paleoclimatology/tree-ring). Green triangles indicate sites with tree ring data (map source: https://www.ncei.noaa.gov/maps/paleo/?layers=0000000000000001, last access: 14 March 2025), and the red rectangular roughly mark the Jing-Jin-Ji Region, i.e., our study area in China.

You are right that it would be helpful if we could provide some connection between the dryness/ wetness indices with other drought indices. We have added the frequency of each dryness/wetness grade at stations (i.e., the basis of establishing regional dryness/wetness indices) to the notes after Eq. (1) in the section 3.1 (please see below), despite the fact that we are currently unable to further compare the dryness/wetness indices with SPEI due to limitations in data availability. We hope this to some extent can provide information on the dryness/wetness indices that the reader might be interested. Many thanks for the advice.

Section 3.1, Lines 145-150:

"*m*, *n*, *j*, and *k* denote the number of stations reporting very dry (grade 5, frequency: about 10%), dry (grade 4, frequency: 20% to 30%), wet (grade 2, frequency: 20% to 30%), and very wet (grade 1, frequency: about 10%)…Stations with grade 3 (normal years, frequency: 30% to 40%) and no data are excluded in the calculation... The frequency of each dryness/wetness grade is from Chinese Academy of Meteorological Sciences (1981)"

**Reference**

Chinese Academy of Meteorological Sciences: Yearly charts of dryness/wetness in China for the last 500-year period, SinoMaps Press, Beijing, 1981.

Cook, E. R., Seager, R., Kushnir, Y., Briffa, K. R., Büntgen, U., Frank, D., Krusic, P. J., Tegel, W., Van Der Schrier, G., Andreu-Hayles, L., Baillie, M., Baittinger, C., Bleicher, N., Bonde, N., Brown, D., Carrer, M., Cooper, R., Čufar, K., Dittmar, C., Esper, J., Griggs, C., Gunnarson, B., Günther, B., Gutierrez, E., Haneca, K., Helama, S., Herzig, F., Heussner, K.-U., Hofmann, J., Janda, P., Kontic, R., Köse, N., Kyncl, T., Levanič, T., Linderholm, H., Manning, S., Melvin, T. M., Miles, D., Neuwirth, B., Nicolussi, K., Nola, P., Panayotov, M., Popa, I., Rothe, A., Seftigen, K., Seim, A., Svarva, H., Svoboda, M., Thun, T., Timonen, M., Touchan, R., Trotsiuk, V., Trouet, V., Walder, F., Ważny, T., Wilson, R., and Zang, C.: Old World megadroughts and pluvials during the Common Era, Sci. Adv., 1, e1500561, https://doi.org/10.1126/sciadv.1500561, 2015.

*Other issues:*

*Line 65: Studies of current drought impacts often involve large samples across wide areas. The text should specific that historic studies of drought impacts tend to specialize in one region or country, reflecting the geographical and chronological specialization typical of historical research.*

**Response**: We have specified the text in and modified it to: "However, little is known about the similarities and differences between individuals or groups in different socio-environmental contexts when facing the same climatic stimulus (i.e., precipitation deficits), as studies on past drought impacts tend to emphasize place-specific (e.g., a given region or country) experiences due to the geographical and chronological specialization of historical research." (Lines 66-69)

*Line 86–87: "Its contemporary territory is 35.7×104 km2" This does not make sense. Do the authors mean "~357,000 km²"?*

**Response**: Sorry for the typo. It has been corrected to $35.7 \times 10^4$ km$^2$ (Line 91).

*Line 104–105: "The total area is 21.6×104 km2." See previous comment.*

**Response**: Sorry for the typo. It has been corrected to $21.6 \times 10^4$ km$^2$ (Line 110).

*Line 79-80: The phrase "immutable nature" seems to suggest that there won't be any changes in drought responses across time, which is obviously not the case. There are both stable and dynamic elements in drought impacts and responses, as the study has found. The language in section 5.1 should also reflect this finding. Perhaps "unchanging nature" should be "stable elements," and "dynamics" should be "dynamic elements." (At least I think this is what the authors mean.)*

**Response**: Thank you very much for the advice. "Stable element" and "dynamic element" in drought impacts and responses are what we intended to express, but apparently they were poorly worded in the manuscript. Sorry for the inappropriate wording. The phrases "immutable/unchanging nature" and "dynamics" in both main text (Lines 16, 70, 85, 558, 693, and 704) and section headings (Section 5.1 and 5.2) have been replaced by stable elements and dynamic elements, respectively.

*Line 86: I hope that readers of this journal will already be aware that "Germany is situated in the central part of the European continent."*

**Response**: This unnecessary statement has been deleted from the beginning of Section 2.

*Line 314–316: "From spring 1831 to spring 1833, rather dry conditions prevailed 66.67% of the time, with one-third of all abnormally dry seasons also accompanied by heatwaves. Among the nine seasons, 33.33% mentioned ecological impacts described as vegetation damage…" It would be better simply to use "two thirds" and "one third" rather than "66.67%" and "33.33%" to avoid a false impression of precision, particularly when it comes to historical records and inferences. The same goes for other cases where small fractions are expressed as percentages down to two decimal places.*

**Response**: Thanks for the advice on how to avoid a false impression of precision. Accordingly, we have looked through the content on drought impacts in Section 4 (Lines 340-353, 383-394, 413-425, 444-458, 478-492, and 516-529) and changed decimal representations to expressions such as "nearly three quarters" and "four of the five seasons."

*Lines 461–462: The caption for Figure 5 does not make sense and should be rewritten. The phrase "being responded" is not grammatical, and its meaning is unclear.*

**Response**: The caption for Figure 5 has been rewritten as "The documentation of and reaction to drought impacts in Germany and the Jing-Jin-Ji Region. The former is presented as (a) the categories and sub-categories of drought impacts that were mentioned in written documents at different social development stages. The latter is illustrated by (b) the percentage of drought categories or sub-categories to which society responded in all five categories or in all 17 sub-categories during each event." The unclear phrases in Figure 5, such as "being responded", have also been corrected to "Social responses to drought effects & impacts."

*Lines 510–516: This discussion presents the differences between the 1921 drought impacts and 2018 drought impacts mainly in terms of another century of modernization. However, the difference in this case has much to do with political stability and legitimacy, given the precarious political conditions in both China and Germany following WWI. Overall, the study may also understate significance social and economic differences between Germany and China during the 20th century, particularly that the German population was more urbanized.*

**Response**: Thank you very much for the advice. It is true that different political stability and legitimacy also contribute to the differences in drought impacts between the past and present, as political conditions have influences on the possibility and/or effectiveness of certain disaster responses. This point should have been presented in comparing the effects of drought on food security in Germany between the 1920 event and 2018 event, but we missed it from the discussion on lines 510–516 in the original manuscript. For example, apart from early interventions to drought impacts on agricultural production, grain imports to Germany also played an important role in ensuring food security during the 2018 drought, which could not be implemented easily in the 1921 drought due to restrictions and overall tough sociopolitical circumstances after WWI, as well as a lower level of cross-boundary collaboration in the early 20th century compared to nowadays. As for social and economic differences (e.g., urbanization, agricultural commercialization) between Germany and China during the 20th century, we think that these are reflected to some extent in the different intermediate links (i.e., raising food price vs. famine) in the path from precipitation deficits to food insecurity to social instability between the two study areas. However, after another read of the relevant content, we realized that such differences were only displayed in the form of impact chains but lacked necessary interpretations.

In the revised manuscript, we have reorganized the discussion on food security to cover the abovementioned missing points and explanations. Please see Lines 627-663 in the Section 5.2 for details on the modification. We hope that the addition of information on socioeconomic conditions in our discussion, which have been briefly presented at the beginning of the analysis on each event in the section 4, would facilitate a more comprehensive and thorough comparison of drought effects on food security across regions and over time.

**Reply to comments from Reviewer #2**

We kindly thank the reviewer for the constructive suggestions and helpful comments on our manuscript, which greatly help the clarification and improvement of event selection, impacts categorization, and so on. Please see below for point-by-point responses. In the reply, comments are quoted in *italics*, and our responses are interleaved in regular font. Meanwhile, line numbers, sections, figures, tables or supplementary materials indicated herein refer to the revised version of manuscript, and they are marked green when pointing out the location of specific modifications in the revised manuscript.

In addition, according to the comments from Reviewer 1, the ambiguous expression "drought manifestations" has been amended to clearer phrases such as "specific effects and impacts of drought" in the revised manuscript and supplement materials. However, we continue to use the original expression "manifestations" in this document when responding to relevant comments, in order to make it easier to match our responses with the original version of our manuscript.

*This is a well-structured and substantial manuscript. Through a highly comprehensive framework, the author compared six drought cases across Germany and China's Beijing-Tianjin-Hebei region, spanning agricultural,*

*industrial, and modern eras. The integration of multi-source data and establishment of a unified analytical framework are two key achievements of this work. They made the comparative analysis reasonable and conclusions credible. The revelation of unchanging nature of drought holds significant value for understanding and better addressing drought challenges.*

**Response**: We appreciate your positive comments on our study and suggestions on how to improve this manuscript.

*Some suggestion:*
 *(1) What is the orange boundary in Figure 1(b)? It seems not right if it represents the boundary of the North China Plain.*

**Response**: Sorry for the mistake. We originally intended to that the Jing-Jin-Ji Region is located in the northern part of North China, a densely populated and drought-prone region of China. We have corrected this mistake in both Figure 1(b) and the main text (Section 2, Lines 110-111).

*(2) Section 3.3 - I noticed that in the impact categories, water deficit or deterioration involved multiple categories. How to identify them in different materials? Although detailed theoretical classifications are provided, it might be also important to clarify how these categories were recorded, described, or quantified across historical, early modern, and modern source materials.*

**Response**: Manifestations relating to the deficit or deterioration of water include:
a)  Precipitation deficits in the meteorological category. It refers to the lack of precipitation (e.g., rainfall, snow, etc.).
b)  Deteriorated water quality in the sub-category of freshwater system in the ecological category. It refers to the deterioration of water quality in natural or artificial water bodies due to drought and/or heat, such as algae bloom, increased pollutant concentration, and oxygen depletion.
c)  Manifestations of drought affecting rivers, lakes/reservoirs, and groundwater in the hydrological category. They denote reduced streamflow and/or water storage due to low water supply after months of inadequate precipitation.
d)  Manifestations in the sub-category of water security in the socioeconomic category. In term of deficit, it refers to insufficient/limited/no water supply for certain purposes, such as drinking, washing and cleaning, watering and irrigation, industrial production, etc. As for deterioration, it mainly concerned with the quality of drinking water; for example, water provided for drinking during droughts was no longer guaranteed to meet drinkable standards.

An additional list is provided at the end of this document and uploaded as a new supplementary material (Supplement B) of this article, which includes keywords for the 56 manifestations in five drought categories, as well as examples of how these manifestations were documented in historical, early modern, and modern (contemporary) source materials. We hope it could provide required details to anyone interested and help the reader to understand our framework.

*(3) Section 3.4 - I totally understand that defining the start and end of meteorological-level precipitation deficits is relatively straightforward and. However, as I know, historical textual records often reflect lagged effects of drought impacts. How do the authors view the omission of such lagged information, and to what extent might this affect the conclusions? A brief clarification would be helpful.*

**Response**: Thank you very much for your kind understanding. Our definition of the event duration is actually a tradeoff between the limited availability and relatively narrow scope of drought records in history (e.g., insufficient soil moisture was mentioned only in severe cases like "cracks in land") and the abundant and comprehensive drought reports at present (e.g., varying degrees of soil moisture at different soil depths). In other words, good documentation of meteorological-level precipitation deficits could be seen as the greatest common denominator of drought narratives across regions and over time.

We acknowledge that two types of lagged information about drought impacts might be missing in the current study due to the event duration we defined, which are as follows:

i.   The first are some cascading effects of drought that take a relatively long time (e.g., more than a year) to develop or be tangible. In order to approximately assess the potential of such omission, we looked again at drought-related descriptions in chronicle and compilation, yearbook, and secondary research (i.e., written documents that were normally composed in the form of after-the-fact summaries) adopted in this study. We found that impacts still reported after sufficient rainfall were more or less a continuation or repetition of previously reported impacts rather than entirely new ones, except for "the harvests of 1922 and 1923 were good, but it was hard to hire agricultural workers because so many had gone to the cities (Li, 2007)" (i.e., potential for affecting crop harvests) shortly after the Jing-Jin-Ji 1920 drought. We believe such omission would not invalidate the main findings of our categorization-based analyses of drought consequences. Nevertheless, we are not saying that these lagged effects are worthless or that such omission hardly raises uncertainties.

ii.  The second are less tangible but far-reaching drought impacts, such as altering social resilience and even accelerating social collapse. This topic is also important for understanding drought challenges, but it somewhat go beyond the focus (or rather capability) of case studies, i.e., to create relatively comprehensive case profiles, analyze single cases in depth, and/or compare a certain issue in different cases.

We fully agree that it is helpful and necessary to make a brief clarification on limitations of this event duration definition. Thus, modification has been made to Lines 280-289 in the Section 3.4 as below:

"This study took the onset and cessation of precipitation deficits as the beginning and end of an event, which was a trade-off between the limited availability and relatively narrow scope of drought records in history and the abundant and comprehensive drought reports at present. As sequels to the lack of precipitation, other effects and impacts of drought may not take place (e.g., electricity shortage) or be well documented (e.g., degree of soil moisture at different soil depths) in every region and at every episode of social development. In addition, some impacts can continue in an inconspicuous manner over a non-measurable span after precipitation returned to normal (e.g., reduced coping capacity for future hazards), which goes more or less beyond the scope of the current case study. Nevertheless, it should be noticed that this definition of event duration is not ideal for exploring drought

effects in the long run (e.g., post-drought labor shortages due to climate exodus) or drawing a complete picture of drought resilience over a period."

*(4) Figure 6- What are the specific criteria for identifying "heat-related manifestations"*

**Response**: For a given manifestation in the socioeconomic category, whether it was related to heat or not was identified based on specific textual information in written documents. A manifestation was considered heat-related when it was clearly documented as being exacerbated or induced by high temperature. We have added necessary explanations in the caption for Figure 6 as below:

"Heat-related impacts refer to those clearly described in written documents as being associated with or caused by abnormal hot conditions, such as increasing imbalances in water or electricity supply and demand due to increased consumption in hot weather, heat-related illness and death, and heat damage to infrastructure (e.g., railway, road)."

*I found that the results and discussion emphasize the focus on concurrent high temperatures during droughts (as reflected in the conclusive title of Section 4.3 and the analytical discussion in Section 5.3). I think that why authors caring about high temperature would be better explained in the main text rather than being confined to table notes (Table 1).*

**Response**: Many thanks for the advice. We have enriched the explanation of why we should care about concurrent high temperatures during droughts and moved it from table notes (Table 1) into the main text in the Section 3.3 (Lines 229-232) as below:

"In addition, compound heatwaves were incorporated into the meteorological category, considering not only the contribution of abnormal high temperature to the likelihood and destructiveness of drought (Zscheischler et al., 2018), but also the not rare descriptions in written documents of the combined effects of heat and drought."

**Reference**
Zscheischler, J., Westra, S., Van Den Hurk, B. J. J. M., Seneviratne, S. I., Ward, P. J., Pitman, A., AghaKouchak, A., Bresch, D. N., Leonard, M., Wahl, T., and Zhang, X.: Future climate risk from compound events, Nature Clim Change, 8, 469–477, https://doi.org/10.1038/s41558-018-0156-3, 2018.

*(5) The current title seems to emphasize regional differences. I think it may be more clear to explicitly highlight the study's scope and conclusions in the title. For example: Cross-era Case Comparison Between Germany and Jing-Jin-Ji Region (China) reveals Invariance and Changes in Drought Impacts and social Responses.*

**Response**: Many thanks for the advice and example of article title. We agree that it would make the title much clearer if we could point out the scope and main conclusions of this study directly. To highlight our research focus, we have changed the article title to "Stable yet dynamic: A cross-era comparative case study of drought impacts and social responses in Germany and Jing-Jin-Ji Region (China)".

**Reply to comments from Community #1**

We appreciate the inspired comments on our manuscript, which not only help strengthen the current comparative analysis but also suggest a potential direction for further improving the impact-response framework. Please see below for point-by-point responses. In the reply, comments are quoted in *italics*, and our responses are interleaved in regular font. Meanwhile, line numbers, sections, figures, tables or supplementary materials indicated herein refer to the revised version of manuscript, and they are marked green when pointing out the location of specific modifications in the revised manuscript.

In addition, according to the comments from Reviewer 1, the ambiguous expression "drought manifestations" has been amended to clearer phrases such as "specific effects and impacts of drought" in the revised manuscript and supplement materials. However, we continue to use the original expression "manifestations" in this document when responding to relevant comments, in order to make it easier to match our responses with the original version of our manuscript.

*The author's comparative study on historical disasters in China and Germany holds significant value, as historical experiences transcending temporal and spatial constraints provide valuable references for current and future human responses to climate change. The indicator system constructed in this paper also offers enlightening implications for researchers seeking to achieve quantitative comparative analyses across different temporal and spatial contexts.*

**Response**: Thank you very much for the positive comments on our study, as well as for pointing out the issues that require our attention and consideration.

*Some issues requiring the author's consideration:*

*1. Quantifying the social impacts and societal responses to droughts: The current classification-based approach lacks quantitative indicators for evaluating impact intensity and response magnitude, which may somewhat compromise the effectiveness of comparative research.*

**Response**: Many thanks for the advice. We agree that quantification would generally strengthen comparative research and make results more straightforward for readers. We also thought of the following two ways to quantify drought impacts and social responses when developing the framework. However, after considering the potential risks and feasibility of quantification in the case of this study, we eventually adopted a strategy of categorization to provide a comprehensive and comparable profile of each drought event, rather than building a quantitative indicator system though methods such as the number/frequency of mentions and gradation. Please see below for our considerations:

**(1) Number/frequency of mentions**

   This method aims to characterize the occurrence of hazards/disasters, the intensity of impacts, or the magnitude of responses based on the absolute number of mentions in written documents during a given period, such as the peasant uprising sequence in China from 210 BC to 1910 AD (Fang et al., 2015), or a proxy indicator

involving multiple dimensions of particular events, such as annual famine index series in the Qing Dynasty according to the amount of famine-affected counties and the famine severity-based weighted index (Wei, 2020; Xiao, 2020). Minimizing the effects of the uneven spatiotemporal distribution and absence of records is crucial to improving the robustness of the method, which often requires a large number of records as statistical samples. In sight of this, Fang et al. (2019, pp.72) suggested that this method should be used for reconstructing sequences of historical climate changes and their impacts only when records are abundant and study areas are relatively large.

Nevertheless, this method has been used to a limited extent in cases studies on past climate-society interactions, as represented by Chen et al. (2024). In their research on the Dingwu Great Famine (1876–1879) in China, the number of drought reports in Shun Pao (申报) was adopted as an indicator of drought impacts on society, and the number of response record in Qingshilu (清实录) was used to divide the early, frequent, and decline phases of social responses.

Single-source data and short time span of research are prerequisites for the practicality of this method in Chen et al. (2024), which, in the case of our study, are not achievable. Since our study covered drought cases across different countries and centuries, it was inevitable to adopt data from multiple sources and in different forms. This brings non-negligible risks to employing the quantification method based on the number/frequency of mentions. On one hand, different forms of written documents usually show differences in information densities and preferences for recording. For example, official archives (e.g., local gazetteers, reports, yearbooks) tend to summarize the most important events in a limited space, resulting in relatively small amount of mentions in spite of considerable drought impacts on socioeconomic systems. While newspapers often follow or even repeat stories, generating much greater mentions of social impacts, especially when multiple modern newspapers are involved. On the other hand, secondary research by historians had to be adopted in our study as a supplement when the original materials were unavailable. However, there are often more than one historical records behind a single piece of information provided in such secondary research, which also restrict the feasibility of counting mentions to quantify impact intensity or response magnitude.

**(2) Gradation based on semantic differential**
This method is designed to convert qualitative narratives into quantitative grades by distinguishing different levels with opposite (e.g., good/bad, strong/weak) and additional (e.g., slightly, very) adjectives. It has been widely used for the documentary-based reconstruction of long-term series of past climate (e.g., temperature, precipitation, and dryness/wetness) and socioeconomic conditions (e.g., agriculture, population, and economy) (Fang et al., 2019).

Collecting words related to a specific topic and identifying their correspondence to different climatic or socioeconomic conditions are necessary steps before developing appropriate grading criteria, which require research from a relatively long-term perspective. These steps are difficult to achieve with only a few case studies, or rather, it is difficult to guarantee the rationality and objectivity of grading criteria based on the current six cases in our study. We have noticed that several case studies focusing on a particular aspect of climate-society interactions used the semantic differential approach to grade social impacts, but their grading

criteria were usually not directly extracted from the records collected for the study cases but grounded in long-term research with a similar focus. For example, in the case study of Chongzhen drought (1627–1644) in China and its impacts on famine, Chen et al. (2024) classified famine into three levels based on descriptions indicating different severity of famine, but the criteria for famine classification were adapted from previous studies on climate and famine in North China during 1736–1911 (Xiao, 2020) and in Jiangsu-Shanghai region during 1644–1911 (Wei, 2020). Unfortunately, for the over 20 manifestations of social impacts covered in our study alone, the subject-specific corpora of the majority of them are still lacking or at least unavailable to us. While creating corpora for different periods, document types, and languages goes somewhat beyond the focus of our current event-based study.

In addition, it might also be a bit tricky when assessing the magnitude of social response, as different standards and indicators could sometimes lead to contradictory results. Taking the 1920 and 1997 drought events in China as examples, if the level of involvement serves as the assessment standard, response magnitude during the 1920 event would be larger than that during the 1997 event because of broader international participations and more types of actors; however, if the assessment focused more on the amount and effectiveness of social responses, the magnitude in the 1997 event would be larger than that in the 1920 event due to about threefold responding ways and less life-threatening manifestations (e.g., famine, drought-induced death). Perhaps, it would be better to quantify response magnitude in studies focusing on particular attributes of social response, as opposed to providing comprehensive and comparable profiles of different drought cases, as in our study. Nonetheless, we recognize the value and necessity of quantitative work. We would like to make long-term efforts to quantify our indicator system in the future.

**Reference**

Chen, X., Tao, L., Tian, F., Su, Y., Pan, J., Chen, S., and Zhai, X.: The Qing's central government response to the most severe drought over the past 300 years, Climatic Change, 177, 108, https://doi.org/10.1007/s10584-024-03767-6, 2024.

Fang, X., Su, Y., Zheng, J., Xiao, L., Wei, Z., and Yin, J.: The Social Impacts of Climate Change in China over the Past 2000 Years, Science Press, Beijing, 2019. [历史气候变化对中国社会经济的影响]

Wei, Z.: Spatio-temporal characteristics of famine and its environmental causes in the Jiangsu-Shanghai region during the Qing Dynasty (1644-1911), Prog. Geogr., 39, 1333–1344, 2020. [清代苏沪地区饥荒的时空变化及其环境因素]

Xiao, L.: Spatio-temporal distribution of famine and its relationshipwith climate, disaster, harvest in North China during 1736–1911, Adv. Earth Sci., 35, 478–487, 2020. [1736—1911 年华北饥荒的时空分布及其与气候、灾害、收成的关系]

*2. Concerning case selection criteria: While the first two case pairs are temporally close (1832/1834, 1920/1921), the temporal distance between the last pair is notably greater (1997/2018). The rationale for this disparity requires explicit clarification.*

**Response**: We have added a clarification of such disparity to the Section 3.2 (Lines 207-211), where selection criteria and results of drought events are given. The clarification is as follows:

"Admittedly, the temporal distance of the two modern drought events was greater than that of the selected paired drought events during the period of 1800–1945, as the dry and wet conditions in the two study areas showed different trends, especially in the past three decades (Fig. 2b). In this situation, priority was given to the severity

and recentness of dry conditions when extremity, modernity, and simultaneity were too difficult to achieve at the same time."

*3. On the classification of historical periods: The tripartite division into agrarian societies, industrialization, and recent years appears overly simplistic. More nuanced differentiation is needed, particularly considering China and Germany's distinct developmental stages within these broad categories. For example, the industrialization level of the North China Plain in 1920 was remarkably low, bearing closer resemblance to that of Germany in 1834 rather than 1921 Germany.*

**Response**: Our consideration of the 1920 drought in China as paired with the 1921 rather 1834 drought in Germany was based on the development of railway system in both study areas. For Germany, its first steam-hauled railway (i.e., Bavarian Ludwig Railway) was constructed in 1835.12, and the rapid expansion of its railway system took place after the establishment of the German Empire in 1871. As for North China, or rather the Jing-Jin-Ji Region, its first steam-hauled railway (Tangxu Railway) was constructed in 1881.11, and its railway system had largely enhanced its accessibility to, at least, Northeast China, Northwest China, and the middle and lower Yangtze River basin by 1920 (American National Red Cross, 1921).

Railway is seen as the key technology that deeply intertwined with the spread of industrialization (Berger, 2019) and is more time-efficient and drought-resilient than water-borne transport, such as the time-honored canal system in China. The influence of the railway system on disaster relief has been recognized, although its impact on the local economic development remains complex. For example, Li (2007) mentioned in her research on the 1920 drought and famine in North China that "there were ample supplies available from Manchuria that could be transported by rail: the Peking-Kalgan, Peking-Mukden, Peking-Hankow, and Tientsin-Pukow lines... The railways were seen as the main factor in limiting the loss of life… For North China as a whole, the estimated mortality was half a million victims, a terrible human toll, but far less than the estimated 9–13 million victims of the 1876–79 famine." Thus, it is rational to assume that the development of railway system might bring about changes in the drought-society interactions before launching into the analysis of specific cases.

Nevertheless, we fully agree with your comment that "the industrialization level of the North China Plain in 1920 was remarkably low", and, in terms of urbanization and livelihoods, this region in 1920 was "closer resemblance to that of Germany in 1834 rather than 1921 Germany." Correspondingly, additional explanations have been added in Section 3.2, where the selection of drought events was introduced, and in Section 4.2, where the 1920 event in China was analyzed. The former provides details about our consideration of period division, while the latter emphasizes the fact that the Jing-Jin-Ji Region was at the early stage of industrialization and was still dominated by the peasant economy. Please see below for details on these additional explanations.

Section 3.2, Lines 211-214:
"It should also be noted that the relatively well-developed railway system was an important consideration for dividing the Jing-Jin-Ji 1920 event into the era of industrialization, although the broad population shift from agriculture to manufacturing had not yet been observed in this region at the time (Li, 2007; Wu, 1985)."

"At that time, this region was witnessing the early stage of industrialization, which brought about the development of railways, but much of it was still under a self-contained natural economy where grains were produced mainly for producers' direct consumption."

**Reference**

American National Red Cross: Report of the China famine relief, American Red Cross, October, 1920-September, 1921., 1921.

Berger, T.: Railroads and Rural Industrialization: evidence from a Historical Policy Experiment, Explorations in Economic History, 74, 101277, https://doi.org/10.1016/j.eeh.2019.06.002, 2019.

Li, L. M.: Fighting famine in North China: State, market, and environmental decline, 1690s~1990s, Standford University Press, Stanford, 2007.

Wu, C.: Chinese capitalism and domestic markets, Chine Social Sciences Press, Beijing, 1985. [中国资本主义与国内市场]

*4. Comparative analysis refinement: The current conclusions lack sufficient clarity in pattern extraction. Enhanced focus should be placed on: (1) Contrasting response mechanisms and their effectiveness between agricultural and industrial eras; (2) Systematic comparison of similarities, differences, and underlying causes in societal response patterns between China and Germany.*

**Response**: Thank you very much for the constructive comment. Following your suggestions, we have added a summary of social response pattern at the end of the analysis of each event in the Section 4 (Lines 370-373, 404-411, 436-442, 468-476, 505-514, and 544-552). Meanwhile, the Conclusion section has been reorganized with focuses on both cross-eras and cross-regional comparisons of response mechanisms, effectiveness and possible causes (Lines 704-758). Please see the abovementioned contents (in green) in the revised manuscript for details on relevant modifications.

---

## Author Response (AR2)

**Reply to comments from Reviewer #1**

We appreciate the suggestions for clear language and appropriate expression, which have indeed further improved the presentation quality of this manuscript. Please see below for point-by-point responses. In the reply, comments are quoted in *italics*, and our responses are interleaved in regular font. In addition, line numbers, sections, figures, tables or supplementary materials indicated herein refer to the revised version of manuscript, and they are marked green when pointing out the location of specific modifications in the revised manuscript.

*This extensive revision of the manuscript addresses the key issues in the first version. The author(s) should address the following points before publication:*
*-The language of the manuscript has been much improved but would still benefit from further review for clarity and correct use of language. In particular, the manuscript still uses some erroneous phrases (e.g. "river cutoff" in line 390) that won't be detected automatically but that don't quite make sense in English.*

**Response**: Thanks for pointing out these erroneous phrases. We have checked the manuscript and modified "river cutoff" to "rivers cease to flow" or "the cessation of river flow" according to the context in line 390, Table 1, and supplementary materials B, C, and D.

*-In line 187, the meaning of the following sentence—and thus the authors' response to the challenge of rapidly changing social conditions within the time-period under study— remains unclear: "This hints at the need for developing selection criteria segmentally."*

**Response**: This unclear sentence has been modified to "The rapid improvement in social conditions and increasingly visible results of climate change distinguish the post-1945 period from earlier time periods under study." Please see line 187 for details.

*-The language of the manuscript needs to reflect the limitations of the available documentary record. In particular, for the earlier period of events, absence of evidence does not constitute conclusive evidence of absence. For example, on lines 371-372, the sentence "Private sector was the only actor not acting in*

*this event" should be something like "The available documentary record provides evidence of responses from all categories of actors except the private sector."*

**Response**: We fully agree that it is necessary to reflect the limitations of the available documentary record. Accordingly, the statement regarding private sector has been modified to "Private sector was the only actor that the available documentary records did not provide evidence of responses" (lines 371-372). In addition, we have specifically rechecked the content related to the earlier period of events and made a modification to lines 434-435 as follows: "There were only two production-level responses mentioned in the available documentary records..."

*-In lines 413-414, "was also limited by the burdens of the First World War" makes it seem that the war was still ongoing. Perhaps it would be better to say that "the economy and state were burdened by the damage of the First War World as well as post-war debt and reparations payments."*

**Response**: Many thanks for the suggestion. We have changed the statement to "At the time, the country had benefited from the technological progress of the Industrial Revolution, but its economy and state were burdened by the damage of the First World War as well as post-war debt and reparations payments…" Please see lines 412-415 for details.

*-In line 450, the meaning of the phrase "with only one sixth of such dry conditions accompanied by heatwaves" is unclear.*

**Response**: The unclear phrase has been modified to "Of the seven seasons, six were identified as abnormal dry seasons, only one of which was accompanied by heatwaves..." (lines 451-452).

*-Section 5.2 should mention the possibility that the greater availability of evidence since the 20th century contributes to the great number and variety of recorded impacts and responses, even if the authors do not believe this is the principal reason for the growth and diversification of recorded impacts and responses.*

**Response**: Many thanks for the suggestion. Additional statement about the changing availability of evidence has been added to lines 607-609 as follows: "In addition to the greater availability of documentary evidence since the 20th century, such changes may also be attributed to two causes, namely increasingly complicated economic sectors and changing social concerns."